# Predictive modeling reveals that higher-order cooperativity drives transcriptional repression in a synthetic developmental enhancer

Yang Joon Kim[1], Kaitlin Rhee[2], Jonathan Liu[3], Selene Jeammet[4], Meghan A Turner[5], Stephen J Small[6], Hernan G Garcia[1,3,5,7,8]*

[1]Chan Zuckerberg Biohub, San Francisco, United States; [2]Department of Chemical Biology, University of California, Berkeley, Berkeley, United States; [3]Department of Physics, University of California, Berkeley, Berkeley, United States; [4]Department of Biology, Ecole Polytechnique, Paris, France; [5]Biophysics Graduate Group, University of California, Berkeley, Berkeley, United States; [6]Department of Biology, New York University, New York, United States; [7]Department of Molecular and Cell Biology, University of California, Berkeley, Berkeley, United States; [8]Institute for Quantitative Biosciences–QB3, University of California at Berkeley, Berkeley, United States

*For correspondence:
hggarcia@berkeley.edu

Competing interest: The authors declare that no competing interests exist.

**Abstract** A challenge in quantitative biology is to predict output patterns of gene expression from knowledge of input transcription factor patterns and from the arrangement of binding sites for these transcription factors on regulatory DNA. We tested whether widespread thermodynamic models could be used to infer parameters describing simple regulatory architectures that inform parameter-free predictions of more complex enhancers in the context of transcriptional repression by Runt in the early fruit fly embryo. By modulating the number and placement of Runt binding sites within an enhancer, and quantifying the resulting transcriptional activity using live imaging, we discovered that thermodynamic models call for higher-order cooperativity between multiple molecular players. This higher-order cooperativity captures the combinatorial complexity underlying eukaryotic transcriptional regulation and cannot be determined from simpler regulatory architectures, highlighting the challenges in reaching a predictive understanding of transcriptional regulation in eukaryotes and calling for approaches that quantitatively dissect their molecular nature.

## Editor's evaluation

The work by Kim et al., used synthetic constructs in *Drosophila* to examine the relationship between regulators and transcription initiation. By measuring regulator concentrations and the corresponding RNA polymerase initiation rates in different synthetic constructs and using a thermodynamic model, the authors concluded that higher-order cooperativities between the repressor on adjacent binding sites, and that between the repressor and RNA polymerase are needed to explain the observed response curves in RNA polymerase loading rate. This work targets a challenging question in eukaryotic transcription regulation, where higher-order cooperativity between different molecular components, in addition to simple transcription factor binding and unbinding, is often necessary to account for observed promoter behaviors when multiple elements (repressors, mediators, activators) exist.

## Introduction

During embryonic development, transcription factors bind stretches of regulatory DNA termed enhancers to dictate the spatiotemporal dynamics of gene expression patterns that will lay out the future body plan of multicellular organisms (*Spitz and Furlong, 2012*; *Small and Arnosti, 2020*). One of the greatest challenges in quantitative developmental biology is to predict these patterns from knowledge of the number, placement, and affinity of transcription factor binding sites within enhancers. The early embryo of the fruit fly *Drosophila melanogaster* has become one of the main workhorses in this attempt to achieve a predictive understanding of cellular decision-making in development due to its well-characterized gene regulatory network and transcription factor binding motifs, and the ease with which its development can be quantified using live imaging (*Garcia et al., 2020*; *Small and Arnosti, 2020*; *Rivera et al., 2019*).

Predictive understanding calls for the derivation of theoretical models that generate quantitative and experimentally testable predictions. Thermodynamic models based on equilibrium statistical mechanics have emerged as a widespread theoretical framework to achieve this goal (*Ackers et al., 1982*; *Vilar and Leibler, 2003*; *Bolouri and Davidson, 2003*; *Bintu et al., 2005b*; *Bintu et al., 2005a*; *Segal et al., 2008*; *Fakhouri et al., 2010*; *Sayal et al., 2016*; *Phillips et al., 2019*; *Eck et al., 2020*). For instance, over the last decade, a dialogue between these thermodynamic models and experiments demonstrated the capacity to quantitatively predict bacterial transcriptional regulation from knowledge of the DNA regulatory architecture (*He et al., 2010*; *Garcia and Phillips, 2011*; *Brewster et al., 2014*; *Garcia et al., 2012*; *Sepúlveda et al., 2016*).

The predictive power of these models is evident when inferring model parameters from simple regulatory architectures and using those parameters to make parameter-free predictions of more complex architectures (*Boedicker et al., 2013a*; *Boedicker et al., 2013b*, *Razo-Mejia et al., 2018*; *Phillips et al., 2019*). Consider, for example, that RNA polymerase II (RNAP)—which we take as a proxy for the whole basal transcriptional machinery—binds to a promoter with a dissociation constant $K_p$. When RNAP is bound, transcription is initiated at a rate $R$ (*Figure 1A*). In the absence of any regulation, a thermodynamic model will only have $K_p$ and $R$ as its free parameters which can be experimentally determined by, for example, measuring mRNA distributions (*Razo-Mejia et al., 2020*). Now, we assume that the parameters $K_p$ and $R$ inferred in this step do not just enable a fit to the data, but that their values represent physical quantities that remain unaltered as more complex regulatory architectures are iteratively considered. As a result, when we consider the case where a single repressor molecule can bind, our model calls for only two new free parameters: a dissociation constant for repressor to its binding motif $K_r$, and a negative cooperativity between repressor and RNAP, $\omega_{rp}$, that makes the recruitment of RNAP to the DNA less favorable when the repressor is bound to its binding site (*Figure 1B*). Once again, after determining $K_r$ and $\omega_{rp}$ experimentally (*Phillips et al., 2019*), we consider the case where two repressors can bind simultaneously (*Figure 1C*). If the repressors interact with RNAP independently of each other, then our model has no remaining free parameters such that we will have reached complete predictive power. However, protein-protein interactions between repressors could exist or even higher-order interactions giving rise to a repressor-repressor-RNAP ternary complex might be present. This extra complexity would require yet another round of experimentation to quantify these interactions represented by $\omega_{rr}$ and $\omega_{rrp}$ in *Figure 1C*, respectively. Even after quantifying these parameters, predictive power might not be reached if, after adding yet another repressor binding site, a complex between all three repressors and RNAP can be formed (*Figure 1D*).

While protein-protein cooperativity captured by $\omega_{rr}$ has been studied both in bacteria (*Ackers et al., 1982*; *Ptashne and Gann, 2002*) and eukaryotes (*Giniger and Ptashne, 1988*; *Ma et al., 1996*; *Lebrecht et al., 2005*; *Parker et al., 2011*; *Fakhouri et al., 2010*; *Sayal et al., 2016*), the necessity of accounting for higher-order interactions such as those described in our example by the $\omega_{rrp}$ and $\omega_{rrrp}$ terms had only been demonstrated in archeae (*Peeters et al., 2013*) and bacteria (*Dodd et al., 2004*). The need to invoke this higher-order cooperativity in eukaryotes only became apparent in the last few years (*Estrada et al., 2016b*; *Park et al., 2019*; *Biddle et al., 2020*). These higher-order cooperativities might be necessary in order to account for the complex interactions mediated by, for example, the recruitment of co-repressors (*Courey and Jia, 2001*; *Walrad et al., 2011*), mediator complex (*Park et al., 2019*), or any other element of the transcriptional machinery. As a result, while posing a challenge to reaching a parameter-free predictive understanding of transcriptional regulation, higher-order cooperativity provides an

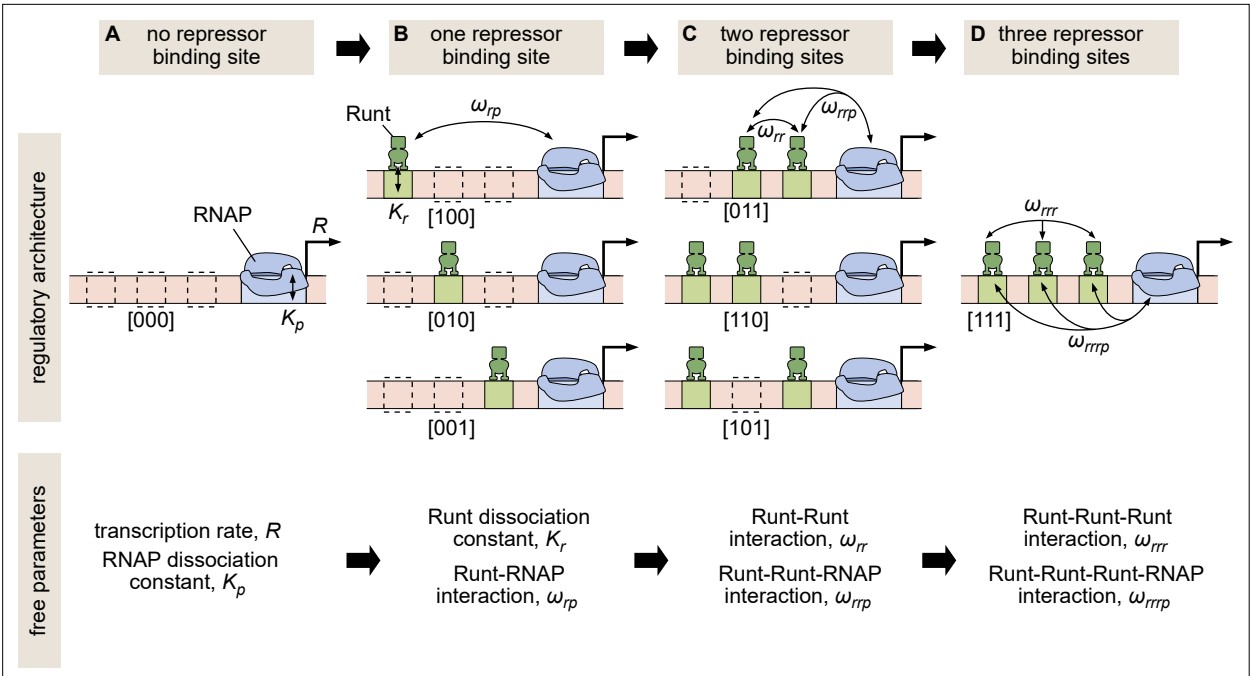

**Figure 1.** Building up predictive models of transcriptional repression. (**A**) In the absence of repressor binding, gene expression can be characterized by a dissociation constant between RNAP and the promoter $K_p$ and the rate of transcription initiation when the promoter is bound by RNAP $R$. (**B**) In the presence of a single repressor binding site, models need to account for two additional parameters describing the repressor dissociation constant $K_r$ and a repressor-RNAP interaction term $\omega_{rp}$. (**C**) For two-repressor architectures, parameters accounting for repressor-repressor interactions $\omega_{rr}$ and for interactions giving rise to a repressor-repressor-RNAP complex could also have to be incorporated. (**D**) For the case of three repressor binding sites, additional parameters $\omega_{rrr}$ and $\omega_{rrrp}$ capturing the higher-order cooperativity between three repressor molecules and between three Runt molecules and RNAP, respectively, could be necessary. Note the nomenclature shown below each construct, which indicates which Runt binding sites are present in each construct.

avenue for quantifying the complexity of the molecular processes underlying eukaryotic cellular decision-making.

In this paper, we sought to test whether an iterative and predictive approach, such as that outlined in *Figure 1*, was possible for transcriptional repression in the early embryo of the fruit fly *Drosophila melanogaster* or whether it is necessary to invoke higher-order cooperativities that challenge the reach of our predictive models as we add more complexity to the system. To make this possible, we engineered binding sites for the Runt repressor into the Bicoid-activated *hunchback* P2 minimal enhancer. We systematically varied the number and placement of Runt binding sites within this enhancer (*Chen et al., 2012*) in order to determine whether model fits to real-time transcriptional measurements from the enhancer constructs containing only one-Runt binding site could accurately predict repression in two- and three-Runt binding site constructs (*Figure 1*). We found that a thermodynamic model can recapitulate all our data. However, we also discovered that, while the model could describe repression by a single Runt repressor, protein-protein and higher-order cooperativities had to be invoked in order to quantitatively account for regulation by two or more repressor molecules. While these higher-order cooperativities limit the iterative bottom-up discourse between theory and experiment that has been successful in bacteria (*Phillips et al., 2009*), they also provide a concrete theoretical framework for quantifying the complexities behind eukaryotic transcriptional control, and call for the development of new theories and experiments specifically conceived to uncover the the molecular underpinnings of this complexity.

## Results

## Predicting transcription rate using a thermodynamic model of Bicoid activation and Runt repression

Inspired by the theory-experiment dialogue leading to predictive understanding of the *lac* operon in *E. coli* over the last four decades (*Phillips et al., 2019*; *Razo-Mejia et al., 2018*; *Garcia and Phillips, 2011*; *Garcia et al., 2012*; *Ackers et al., 1982*; *Buchler et al., 2003*), we built a predictive model of Runt repression on the Bicoid-activated *hunchback* P2 enhancer using the thermodynamic model framework (*Phillips et al., 2019*; *Bintu et al., 2005b*; *Bintu et al., 2005a*) with the goal of predicting the rate of transcription initiation as a function of input transcription factor concentration, and the number and placement of Runt repressor binding sites. Our model rests on

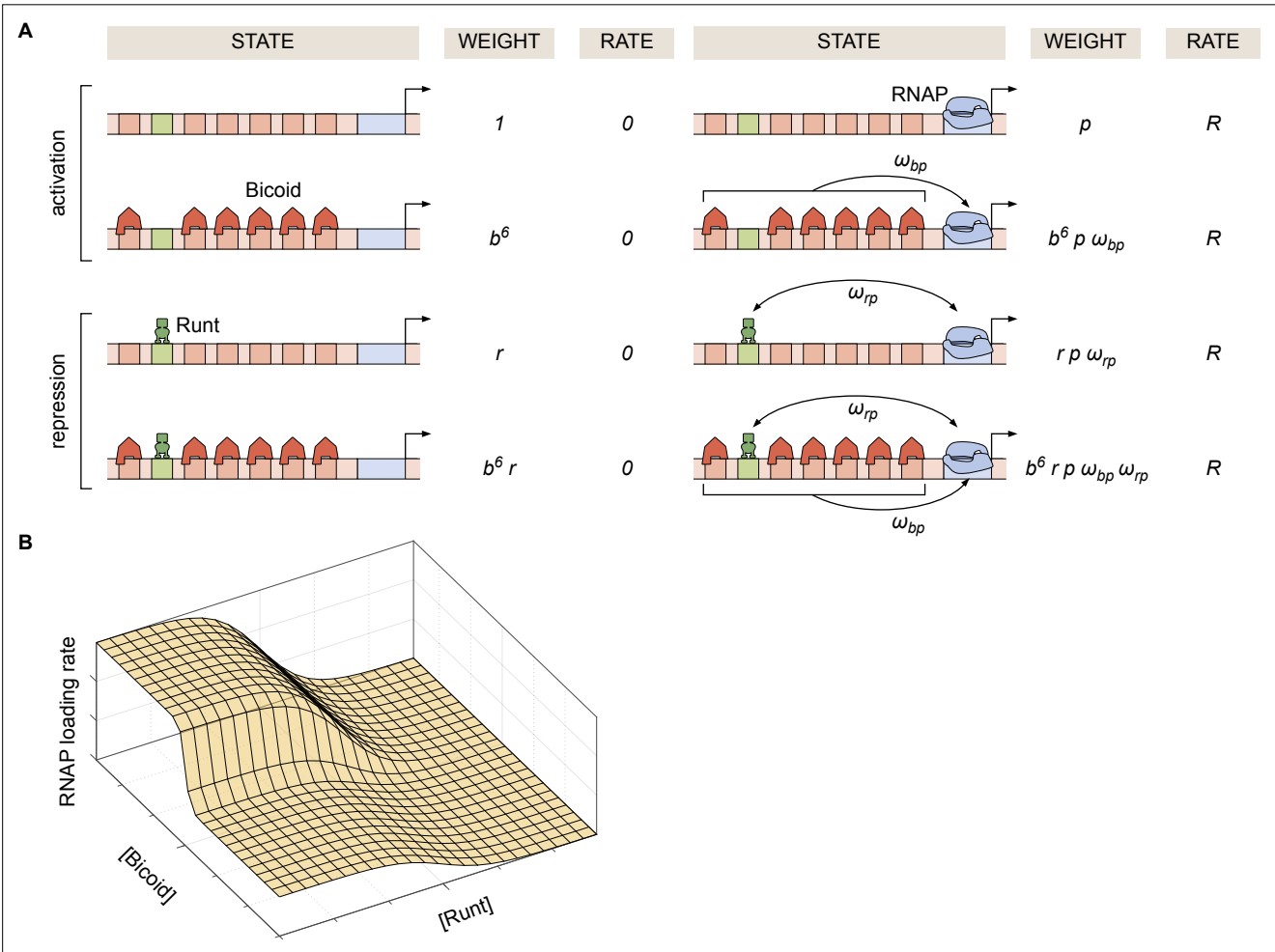

**Figure 2.** Thermodynamic model of transcriptional regulation by Bicoid activator and Runt repressor. (**A**) States and statistical weights for the regulation of *hunchback* P2 with one Runt binding site in the limit of strong Bicoid-Bicoid cooperativity. Here, we use the dimensionless parameters $b = [Bicoid]/K_b$, $r = [Runt]/K_r$, and $p = [RNAP]/K_p$, where $K_b$, $K_r$, and $K_p$ are the dissociation constants of Bicoid, Runt, and RNAP, respectively. $\omega_{bp}$ represents the cooperativity between Bicoid and RNAP, $\omega_{rp}$ captures the cooperativity between Runt and RNAP, and $R$ represents the rate of transcription when the promoter is occupied by RNAP. The top two rows correspond to states where only Bicoid and RNAP act, while the bottom two rows represent repression by Runt. (**B**) Representative prediction of RNAP loading rate as a function of Bicoid and Runt concentrations for $\omega_{bp} = 3, \omega_{rp} = 0.001, p = 0.001, R = 1 (AU/min)$.

The online version of this article includes the following figure supplement(s) for figure 2:

**Figure supplement 1.** General thermodynamic model for a *hunchback* P2 enhancer with six Bicoid binding sites.

**Figure supplement 2.** General thermodynamic model for an enhancer with six-Bicoid binding sites and one Runt binding site.

the 'occupancy hypothesis' that states that the rate of mRNA production, $d[mRNA]/dt$, is proportional to the probability of the promoter being bound by RNA polymerase II (RNAP), $p_{bound}$, such that

$$\frac{d\,[mRNA]}{dt} = R\,p_{bound}, \tag{1}$$

where $R$ is the rate of mRNA production when the promoter is occupied by RNAP. Note that, throughout this study, we treat the rate of transcription initiation and the rate of RNAP loading interchangeably.

To generate intuition, we start by modeling the case of *hunchback* P2 with one Runt binding site. **Figure 2A** illustrates the possible states the system can be found in. Each state has an associated statistical weight which can be calculated as prescribed by equilibrium statistical mechanics (**Bintu et al., 2005b**; **Bintu et al., 2005a**). Here, we assume that there are six Bicoid binding sites with the same dissociation constant given by $K_b$, one Runt binding site with a dissociation constant specified by $K_r$, and a promoter with a dissociation constant for RNAP prescribed by $K_p$. In the absence of Runt, we consider four states as shown in the top two rows of **Figure 2A**. Here, we assume that Bicoid-Bicoid cooperativity is so strong that the enhancer can either be unoccupied or completely bound by Bicoid molecules (**Gregor et al., 2007**; **Park et al., 2019**). Further, we consider an interaction between Bicoid and RNAP given by $\omega_{bp}$. For simplicity, we use the dimensionless parameters $b = [Bicoid]/K_b$, $r = [Runt]/K_r$ and $p = [RNAP]/K_p$. These assumptions lead to a functional form reminiscent of a Hill function that explains the sharp step-like expression pattern along the embryo's anterior-posterior axis of the *hunchback* gene (**Gregor et al., 2007**; **Park et al., 2019**; **Driever and Nüsslein-Volhard, 1988**; **Driever et al., 1989**). A full thermodynamic model in which we do not make this assumption of high Bicoid-Bicoid cooperativity is discussed in detail in Section 'Derivation of the general thermodynamic model for the *hunchback* P2 enhancer' and Section 'Derivation of the general and simpler thermodynamic model for the hunchback P2 enhancer with one Runt binding site'.

The molecular mechanism by which Runt downregulates transcription of its target genes remains unclear (**Chen et al., 2012**; **Hang and Gergen, 2017**; **Koromila and Stathopoulos, 2017**; **Koromila and Stathopoulos, 2019**). Here, we assume the so-called 'direct repression' model (**Gray et al., 1994**) that posits that Runt operates by inhibiting RNAP binding to the promoter through a direct Runt-RNAP interaction term given by $\omega_{rp} < 1$ independently of Bicoid. As a result, in the presence of Runt, we consider four additional states as shown in the bottom two rows of **Figure 2A**. Other potential mechanisms of Runt repression are further discussed in Supplementary Section 'Comparison of different modes of repression', where we also show that the choice of specific mechanism does not change our conclusions.

Given these assumptions, we arrive at the microstates and corresponding statistical weights shown in **Figure 2A**. The probability of finding RNAP bound to the promoter, $p_{bound}$, is calculated by dividing the sum of all statistical weights featuring RNAP by the sum of the weights of all possible microstates. The calculation of $p_{bound}$ combined with **Equation 1** leads to the expression

$$Rate = Rp_{bound} = R\,\frac{p+b^6 p\omega_{bp}+rp\omega_{rp}+b^6 rp\omega_{bp}\omega_{rp}}{1+b^6+r+b^6 r+p+b^6 p\omega_{bp}+rp\omega_{rp}+b^6 rp\omega_{bp}\omega_{rp}}, \tag{2}$$

which makes it possible to predict the output rate of mRNA production as a function of the input concentrations of Bicoid and Runt (**Figure 2B**). With this theoretical framework in hand, we experimentally tested the predictions of this model.

## Measuring transcriptional input-output to test model predictions

The transcriptional input-output function in **Figure 2B** indicates that, in order to predict the rate of RNAP loading and to test our theoretical model, we need to first measure the concentration of the input Bicoid and Runt transcription factors. In order to quantify the concentration profile of Bicoid, we used an established eGFP-Bicoid line (**Gregor et al., 2007**) and measured mean Bicoid nuclear concentration dynamics along the anterior-posterior axis of the embryo over nuclear cycles 13 and 14 (nc13 and nc14, respectively) as shown in Movie **Figure 3—video 1** (**Eck et al., 2020**). An example snapshot and time trace of Bicoid nuclear concentration dynamics at 40% of the embryo length appear in **Figure 3A and B**.

Quantification of the Runt concentration using standard fluorescent protein fusions is not possible due to the slow maturation times of these proteins (**Bothma et al., 2018**). We therefore measured

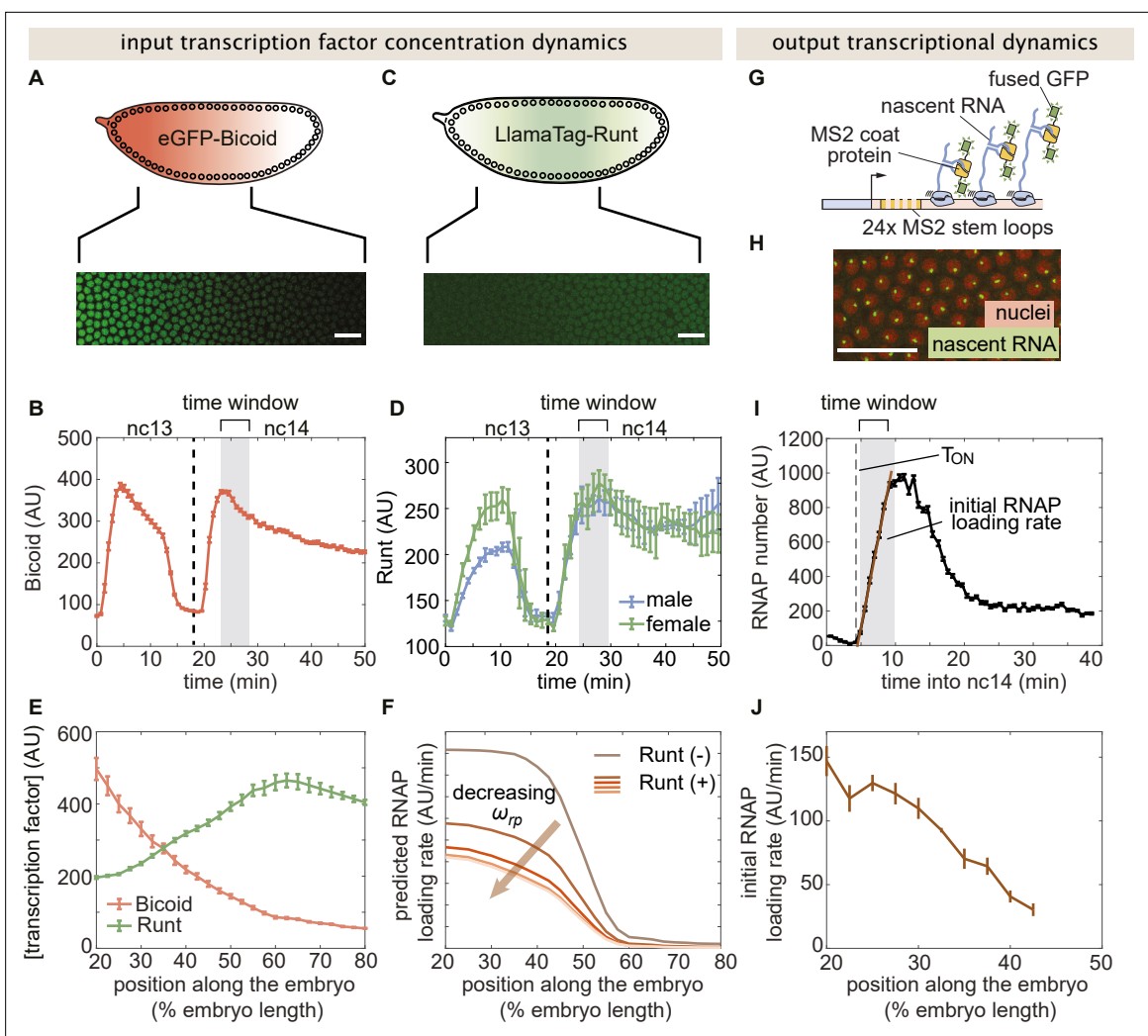

**Figure 3.** Measurement of input transcription factor concentrations and output rate of transcription to test model predictions. (**A**) Snapshot of an embryo expressing eGFP-Bicoid spanning 20–60% of the embryo length. (For a full time-lapse movie, see Movie *Figure 3—video 1*) (**B**) Bicoid nuclear fluorescence dynamics taken at 40% of the embryo. (**C**) Snapshot of an embryo expressing eGFP:LlamaTag-Runt spanning 20–60% of the embryo length. (For a full time-lapse movie, see Movie *Figure 3—video 2*) (**D**) Runt nuclear concentration dynamics in males and females. (**E**) Measured transcription factor concentration profiles along the anterior-posterior axis of the embryo. The concentration profiles are averaged over the gray shaded regions shown in (**B**) and (**D**) which corresponds to a time window between 5 and 10min into nc14. (**F**) Predicted RNAP loading rate for *hunchback* P2 with one Runt binding site over the anterior-posterior axis generated for a reasonable set of model parameters $K_b = 30$ AU, $K_r = 100$ AU, $\omega_{bp} = 100$, $p = 0.001$, and $R = 1$ AU/min for varying values of the Runt-RNAP interaction term $\omega_{rp} = [10^{-2}, 1]$. (**G**) Schematic of the MS2 system where 24 repeats of the MS2 loop sequence are inserted downstream of the promoter followed by the *lacZ* gene. The MS2 coat protein (MCP) fused to GFP binds the MS2 loops. (**H**) Example snapshot of an embryo expressing MCP-GFP and Histone-RFP. Green spots correspond to active transcriptional loci and red circles correspond to nuclei. Spot intensities are proportional to the number of actively transcribing RNAP molecules. (**I**) Representative MS2 fluorescence averaged over a narrow window (2.5% of the embryo length) along the anterior-posterior axis of the embryo. The initial rate of RNAP loading was obtained by fitting a line (brown) to the initial rise of the data and the x-intercept is defined as the onset of transcription ($T_{ON}$). (**J**) Measured initial rate of RNAP loading (over a spatial bin of 2.5% of the embryo length) across the anterior-posterior axis of the embryo, from the *hunchback* P2 enhancer. (B, D, E, and J, error bars represent standard error of the mean over $\geq 3$ embryos; I, error bars represent standard error of the mean over the spatial averaging corresponding to roughly ten nuclei; A, C, and H, white scale bars represent 20 μm.).

The online version of this article includes the following video and figure supplement(s) for figure 3:

**Figure supplement 1.** Comparison of the predicted rate of transcription using dynamic and time-averaged transcription factor concentration profiles as inputs.

**Figure supplement 2.** Initial rate of RNAP loading in nuclear cycle 14 across the anterior-posterior axis for different constructs, with or without Runt protein.

**Figure supplement 3.** Duration of transcription over nuclear cycle 14.

*Figure 3 continued on next page*

*Figure 3 continued*

**Figure supplement 4.** Fraction of competent loci in nuclear cycle 14 along the anterior-posterior axis for each synthetic enhancer construct in the presence and absence of Runt protein.

**Figure supplement 5.** Accumulated mRNA during nuclear cycle 14 along the anterior-posterior axis for each synthetic enhancer construct in the presence and absence of Runt protein.

**Figure supplement 6.** Accumulated mRNA during nuclear cycle 14 versus Runt concentration for each synthetic enhancer construct in the presence and absence of Runt protein.

**Figure supplement 7.** Snapshots of an embryo expressing eGFP:LlamaTag-Runt and the Histone-iRFP signal used for nuclear segmentation.

**Figure supplement 8.** Correlation between the initial RNAP loading rate and accumulated mRNA during nuclear cycle 14.

**Figure 3—video 1.** eGFP-Bicoid confocal movie.

https://elifesciences.org/articles/73395/figures#fig3video1

**Figure 3—video 2.** eGFP:LlamaTag-Runt confocal movie.

https://elifesciences.org/articles/73395/figures#fig3video2

**Figure 3—video 3.** [001]-MS2V5:MCP-GFP (+Runt) confocal movie.

https://elifesciences.org/articles/73395/figures#fig3video3

Runt concentration dynamics using our recently developed LlamaTags, which are devoid of such maturation dynamics artifacts (*Bothma et al., 2018*). Specifically, we generated a new fly line harboring a fusion of a LlamaTag against eGFP to the endogenous *runt* gene using CRISPR/Cas9-mediated homology-directed repair (Materials and Methods; *Harrison et al., 2010*, *Gratz et al., 2015*).

Using this LlamaTag fusion, we measured the mean Runt nuclear fluorescence along the anterior-posterior axis of the embryo over nc13 and nc14 (Materials and Methods; *Figure 3B*; Movie *Figure 3—video 2*). As expected due to the location of the *runt* gene on the X chromosome (*Lott et al., 2011*), there is a sex dependence in the nuclear concentration levels in nc13, with males displaying lower Runt levels than females; this difference is compensated by early nc14 (*Figure 3C and D*). As a result, for ease of analysis, we focused subsequent quantitative dissection on nc14.

We used the measured input protein concentration profiles to predict the output transcription rate. To make this possible, we invoked previous observations stating that the concentration dynamics of input transcription factors does not significantly affect the initial rate of RNAP loading (*Garcia et al., 2013*; *Eck et al., 2020*). As a result, we decided to use the time-averaged concentration dynamics of Bicoid and Runt over a time window spanning 5 min after the 13th anaphase to 10 min after this anaphase (gray shaded region in *Figure 3B and D*) as inputs to our model, resulting in the static spatial concentration profiles shown in *Figure 3E*. We then used these time-averaged concentration profiles of input transcription factors to calculate the time-averaged rate of transcription initiation over the same time window. In the Supplementary Information Section 'Comparing using static versus dynamic transcription factor concentrations as model inputs' we compare this methodology with one that acknowledges input transcription factor concentration dynamics and show that the prediction stemming from both approaches leads to equivalent theoretical predictions. Specifically, the time-averaged rate of transcription predicted by the dynamic inputs was similar to the rate of transcription predicted by the static inputs.

Along the anterior-posterior axis of the embryo, the measured Bicoid and Runt concentration profiles define a trajectory through the input-output function (*Figure 2B*). Given a set of parameters, this trajectory predicts the initial rate of RNAP loading. This quantitative prediction can be directly compared with experimentally measured transcription initiation rates. For example, given the concentration profiles shown in *Figure 3E*, we calculate the RNAP loading rate as a function of the position along the embryo for different values of the Runt-RNAP interaction, captured by $\omega_{rp}$. *Figure 3F* illustrates how $\omega_{rp}$ shapes the predicted profiles for the RNAP loading rate. As expected, the prediction shows that the rate of transcription decreases as the strength of the Runt-RNAP interaction decreases.

Next, we sought to experimentally test these predictions by measuring the rate of RNAP loading using the MS2 system (*Bertrand et al., 1998*; *Lucas et al., 2013*; *Garcia et al., 2013*). Here, we inserted 24 repeats of the MS2 loop sequence following the *hunchback* P2 enhancer and *even-skipped* promoter in our reporter construct, which leads to the fluorescent labeling of sites of active transcription in living embryos (*Figure 3G and H*; Movie *Figure 3—video 3*). The fluorescence intensity of each MS2 spot is proportional to the number of actively transcribing RNAP molecules (*Garcia et al., 2013*).

In order to quantify the transcriptional activity reported by MS2, we measured the mean MS2 spot fluorescence over nuclei in a narrow spatial window (*Figure 3I*; *Garcia et al., 2013*; *Eck et al., 2020*). To measure the initial rate of RNAP loading, we obtained the slope of the initial rise in the number of actively transcribing RNAP molecules over the same time window used to average input transcription factor concentration (*Figure 3I*, brown line). The resulting RNAP loading rate plotted over the anterior-posterior axis is in qualitative agreement with the classic pattern driven by the *hunchback* P2 minimal enhancer (*Figure 3J*; *Garcia et al., 2013*, *Chen et al., 2012*, *Park et al., 2019*).

While we chose the initial rate of transcription as the experimental measurable to confront against our model predictions, the MS2 technique can also report on other dynamical features of transcription such as the time window over which transcription occurs and the fraction of loci that engage in transcription at any point over the nuclear cycle. Although these two quantities have been shown to be relevant in shaping gene expression patterns in other regulatory contexts (*Garcia et al., 2013*; *Lammers et al., 2020*; *Eck et al., 2020*; *Dufourt et al., 2018*; *Reimer et al., 2021*), we found that the transcription time window was not significantly regulated in the presence of Runt (*Figure 3—figure supplement 3*). As described in Section 'Quantitative interpretation of MS2 signals', we did find some modulation of the fraction of transcriptionally engaged loci for a subset of our synthetic enhancer constructs but, as we could not detect a clear trend in how this fraction of active loci was modulated, we did not pursue a theoretical dissection of the control of this quantity by Runt.

## Enhancer sequence dictates unrepressed transcription rates by determining RNAP-promoter interactions

With these theoretical models and our experimental platform in hand, we designed a set of synthetic enhancer constructs with differing number and placement of Runt binding sites as shown in *Figure 4A* (top) , and *Figure 4—figure supplement 1*. Our enhancer sequences are identical to those created and validated by *Chen et al., 2012*, which kept the length of the enhancer sequence consistent and inserted experimentally validated Runt binding sites (*Melnikova et al., 1993*; *Lewis et al., 1999*; *Chen et al., 2012*; *Koromila and Stathopoulos, 2017*) by mutating the base pairs within the enhancer that are not mapped to binding sites for any known transcripiton factor in the early fruit fly embryo (*Hertz et al., 1990*; *Hertz and Stormo, 1999*).

A major assumption of our theoretical approach is that the model parameters obtained from simple regulatory architectures can be used as inputs for more complex constructs. For instance, we assume that the Runt-independent model parameters for Bicoid and RNAP action—$K_b$, $\omega_{bp}$, $p$ and $R$ (*Figure 2A*)—are conserved for all constructs containing Runt binding sites regardless of their number and placement in the enhancer. If model parameters can be shared across constructs, then our model should predict the same profile for the rate of transcription across all synthetic enhancer constructs.

To test this assumption, we measured the initial rate of RNAP loading in all of our reporter constructs, in *runt* null embryos (Materials and Methods). Notably, unrepressed transcription rates varied significantly across synthetic enhancers (*Figure 4A*). For example, despite no Runt being present, the [001] construct had almost twice the unrepressed rate of [000].

This large construct-to-construct variability in unrepressed transcription rates likely originates from the Runt binding site sequences interfering with some combination of Bicoid and RNAP function. To uncover the mechanistic effect of these Runt binding sites sequences on unrepressed activity, we sought to determine which parameters in our thermodynamic model varied across constructs. In the absence of Runt repressor, only four states remain corresponding to the two top rows of *Figure 2A*. In this limit, the predicted rate of transcription is given by

$$Rate = R \frac{\textcolor{red}{p} + \left(\frac{[Bicoid]}{\textcolor{red}{K_b}}\right)^6 \textcolor{red}{p}\textcolor{red}{\omega_{bp}}}{1 + \textcolor{red}{p} + \left(\frac{[Bicoid]}{\textcolor{red}{K_b}}\right)^6 + \left(\frac{[Bicoid]}{\textcolor{red}{K_b}}\right)^6 \textcolor{red}{p}\textcolor{red}{\omega_{bp}}}, \tag{3}$$

where we have invoked the same parameters as in *Figure 2* and *Equation 2*. For clarity, the free parameters in this equation are marked using the red color.

To obtain the model parameters for each construct measured in *Figure 4A*, we invoked the Bayesian inference technique of Markov Chain Monte Carlo (MCMC) sampling that has been widely used for inferring the biophysical parameters from theoretical models (*Liu et al., 2021*, *Razo-Mejia*

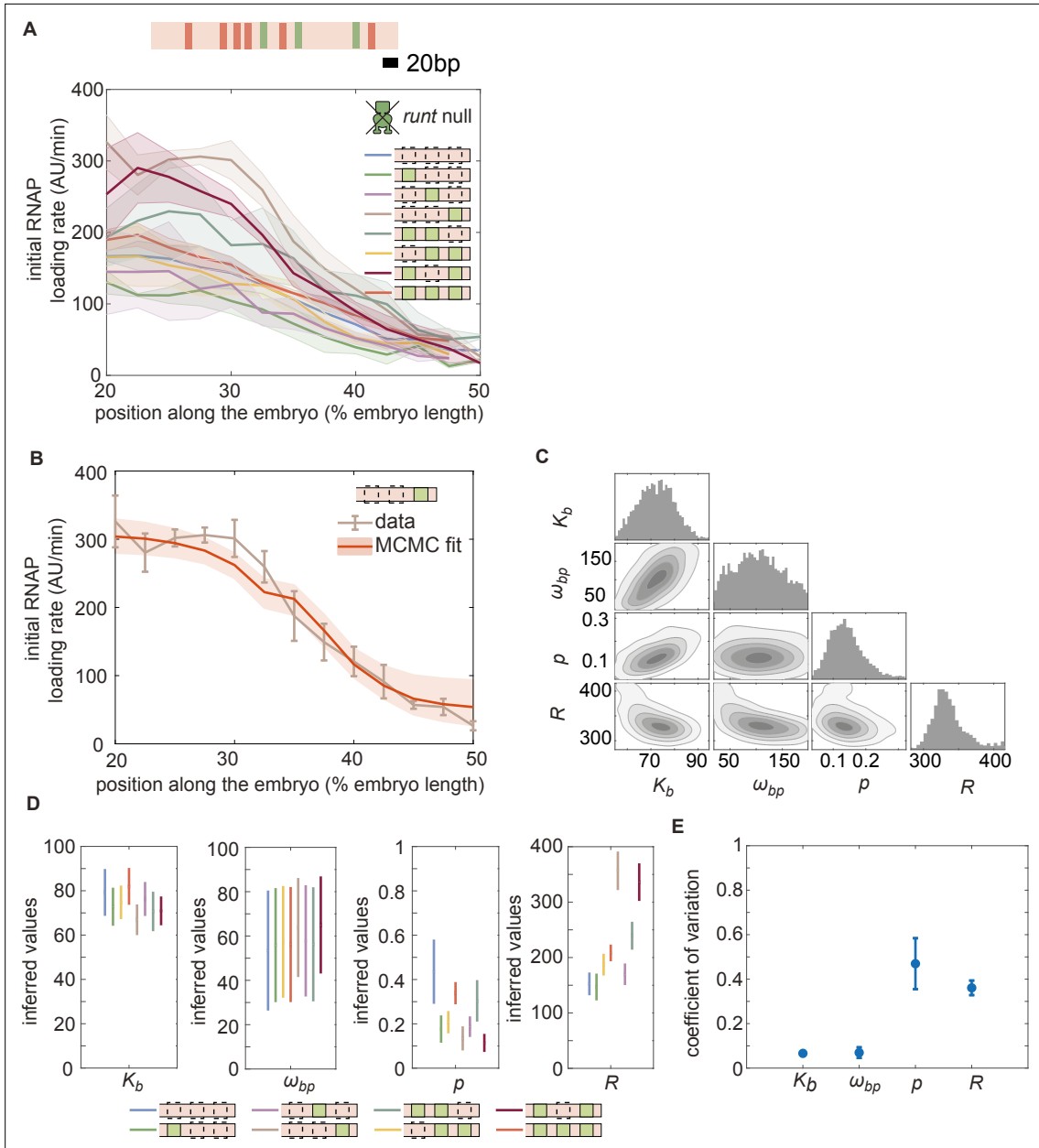

**Figure 4.** Enhancer-to-enhancer variability in the unrepressed transcription level stems from unique RNAP-dependent parameters. (**A**) Measured initial rates of RNAP loading across the anterior-posterior axis of the embryo for all synthetic enhancer constructs in the *absence* of Runt protein. (The [111] synthetic enhancer construct with the position of Bicoid (red) and Runt (green) binding sites is shown in genomic length scale on top as a reference.) (**B**) Representative best MCMC fit and (**C**) associated corner plot for the [001] construct in the *runt* null background. (**D**) Inferred model parameters for all synthetic enhancers in the absence of Runt repressor. Note the large spread in $\omega_{bp}$, consistent with the corner plot shown in (**C**), which indicates that our model does not constrain this parameter well compared to the other parameters. (**E**) Coefficient of variation of inferred parameters. (**A, B**), shaded regions represent the standard error of the mean over>3 embryos; (**B**) error bars from MCMC fit represent 95% confidence interval; (**D**) error bars represent standard deviations calculated from the MCMC posterior chains; (**E**) error bars are calculated by propagating the standard deviation of individual parameters from their MCMC chains.

The online version of this article includes the following figure supplement(s) for figure 4:

**Figure supplement 1.** Bioinformatically predicted architecture of major transcription factor binding sites in the *hunchback* P2 minimal enhancer with three Runt (Run) binding sites.

**Figure supplement 2.** Initial rate of RNAP loading in nuclear cycle 14 across the anterior-posterior axis for different constructs in individual embryos in the absence of Runt protein.

*et al., 2018*, *Geyer and Thompson, 1992*; Supplementary Section 'Markov Chain Monte Carlo inference protocol'). A representative comparison of the MCMC fit to the experimental data reveals a good agreement between theory and experiment (*Figure 4B*). MCMC sampling also gives the distribution of the posterior probability for each parameter as well as their cross-correlation (*Figure 4C*). These corner plots reveal relatively unimodal posterior distributions, suggesting that a unique set of parameters can explain the data.

Note that, while the Bicoid dissociation constant $K_b$ and the Bicoid-RNAP interaction term $\omega_{bp}$ remain largely unchanged regardless of enhancer sequence, there is considerable variability in the inferred mean RNAP-dependent parameters $p$ and $R$ (*Figure 4D*). This variability can be further quantified by examining the coefficient of variation,

$$CV = \frac{\sigma}{\mu}, \tag{4}$$

where $\sigma$ and $\mu$ are the standard deviation and the mean of each parameter, respectively, calculated over all constructs. The coefficients of variation for the RNAP and promoter-dependent parameters are much higher than those for Bicoid-dependent parameters ($\approx 40\%$ versus $< 10\%$; *Figure 4E*). This suggests that the variability in unrepressed transcription rates due to the presence of Runt binding sites stems from differences in the behavior of RNAP at the promoter rather than differences in Bicoid binding or activation. As a result, as we consider increasingly more complex regulatory architectures, we associated each construct with its own specific Bicoid- and RNAP-dependent parameters as inferred in *Figure 4D*. In contrast, as we will show below, we will conserve Runt-dependent parameters as we consider increasingly more complex constructs featuring more Runt binding sites.

## The thermodynamic model recapitulates repression by one Runt binding site

Next, we asked whether our model recapitulates gene expression for the *hunchback* P2 enhancer with a one-Runt binding site in the presence of Runt repressor as predicted by *Equation 2*. We posited that, since the binding site sequence remains unaltered throughout our constructs (*Figure 4—figure supplement 1*), the value of the Runt dissociation constant $K_r$ would also remain unchanged across these enhancers regardless of Runt binding site position; however, we assumed that, as the distance between Runt and the promoter varied, so could the Runt-RNAP interaction term $\omega_{rp}$.

We measured the initial rate of transcription along the embryo for all our constructs containing one Runt binding site in the presence of Runt protein. In this case of a single Runt binding site, *Equation 2* predicts that the initial rate of RNAP loading will be given by

$$Rate = R p_{bound} = R \frac{p + b^6 p\, \omega_{bp} + \frac{[Runt]}{K_r} p\, \omega_{rp} + b^6 \frac{[Runt]}{K_r} p\, \omega_{bp}\omega_{rp}}{1 + b^6 + \frac{[Runt]}{K_r} + b^6 r + p + b^6 p\, \omega_{bp} + \frac{[Runt]}{K_r} p\, \omega_{rp} + b^6 \frac{[Runt]}{K_r} p\, \omega_{bp}\, \omega_{rp}}. \tag{5}$$

Here, we have have rewritten *Equation 2* to clarify which parameters are fixed and which parameters are inferred using color coding. Specifically, we took Runt-independent parameters ($K_b$, $\omega_{bp}$, $p$ and $R$), shown in black, as given by the inference from our previous experiments in the absence of Runt (*Figure 4*). Further, Runt-dependent parameters ($K_r$ and $\omega_{rp}$) which we will infer, are shown in red. We then used MCMC sampling to infer these Runt-dependent parameters for each of our constructs while retaining the mean values of Runt-independent parameters.

The resulting MCMC fits show significant agreement with the experimental data (*Figure 5A*), confirming that, within our model, the same dissociation constant $K_r$ can be used for all Runt binding sites regardless of their position within the enhancer. Further, the corner plot yielded a unimodal distribution of posterior probability of the inferred parameters (*Figure 5B*), indicating the existence of a unique set of most-likely model parameters. We challenged our assumption of constant $K_r$ across our constructs in Section *Figure 5—figure supplement 3*, where we show that, even if we posit that each construct has a different Runt dissociation constant, the obtained $K_r$ values are comparable.

The observed trend in the Runt-RNAP interaction captured by $\omega_{rp}$ qualitatively agrees with the "direct repression" model. Specifically, because the model assumes that Runt interacts directly with RNAP, it predicts that, the farther apart Runt and the promoter are, the lower this interaction should be (*Gray et al., 1994*). In agreement with this prediction, the mean value of $\omega_{rp}$ obtained from our fits changes from high repression ($\omega_{rp} \approx 0.1$) in the [001] construct to almost no repression ($\omega_{rp} \approx 1$) in the [100] construct as the Runt site is moved away from the promoter (*Figure 5C*). Thus, the direct

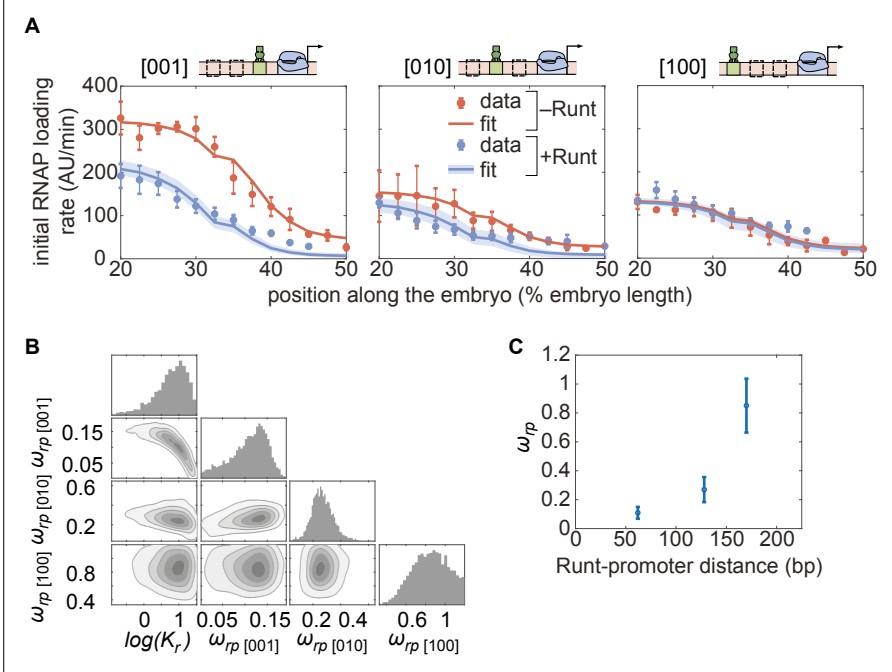

**Figure 5.** Testing the direct repression model in the presence of one Runt binding site. (**A**) Initial transcription rate as a function of position along the embryo for the three constructs containing one Runt binding site in the presence and absence of Runt repressor, together with their best MCMC fits. (**B**) Corner plots from MCMC inference for all constructs with one Runt binding site. (**C**) Inferred $\omega_{rp}$ value as a function of distance between the promoter and the Runt binding site. (A, data points represent mean and standard error of the mean over the embryos and shaded error bars represent 95% confidence intervals for the best MCMC fits for Runt WT datasets; C, data and error bars represent the mean and standard deviation of the posterior chains, respectively.).

The online version of this article includes the following figure supplement(s) for figure 5:

**Figure supplement 1.** Thermodynamic models for different modes of repression.

**Figure supplement 2.** MCMC fitting to the *hunchback* P2 with one Runt binding site constructs using different models of repression.

**Figure supplement 3.** Assessment of alternative models for the one-Runt binding site case.

---

repression model recapitulates repression by a single Runt molecule using the the same dissociation constant regardless of Runt binding site position, and displays the expected dependence of the Runt-RNAP interaction term on the distance between these two molecules.

## Predicting repression by two-Runt binding sites requires both Runt-Runt and Runt-Runt-RNAP higher-order cooperativity

Could the parameters inferred in the preceding section be used to accurately predict repression in the presence of two Runt binding sites? An extra Runt binding site enables new protein-protein interactions between Runt molecules and RNAP (*Figure 6A*). First, we considered individual Runt-RNAP interaction terms, $\omega_{rp1}$ and $\omega_{rp2}$, whose values were already inferred from the one-Runt binding site constructs as $\omega_{rp[001]}$, $\omega_{rp[010]}$, and $\omega_{rp[100]}$ (*Figure 5D*). Second, we considered protein-protein interactions (positive or negative) between two Runt molecules, $\omega_{rr}$. Third, following recent studies of Bicoid activation of the *hunchback* P2 minimal enhancer (*Estrada et al., 2016a*; *Park et al., 2019*), we also posited the existence of simultaneous Runt-Runt-RNAP higher-order cooperativity $\omega_{rrp}$. Given these different cooperativities, and as shown in detail in *Figure 6—figure supplement 6B*, the predicted rate of transcription is

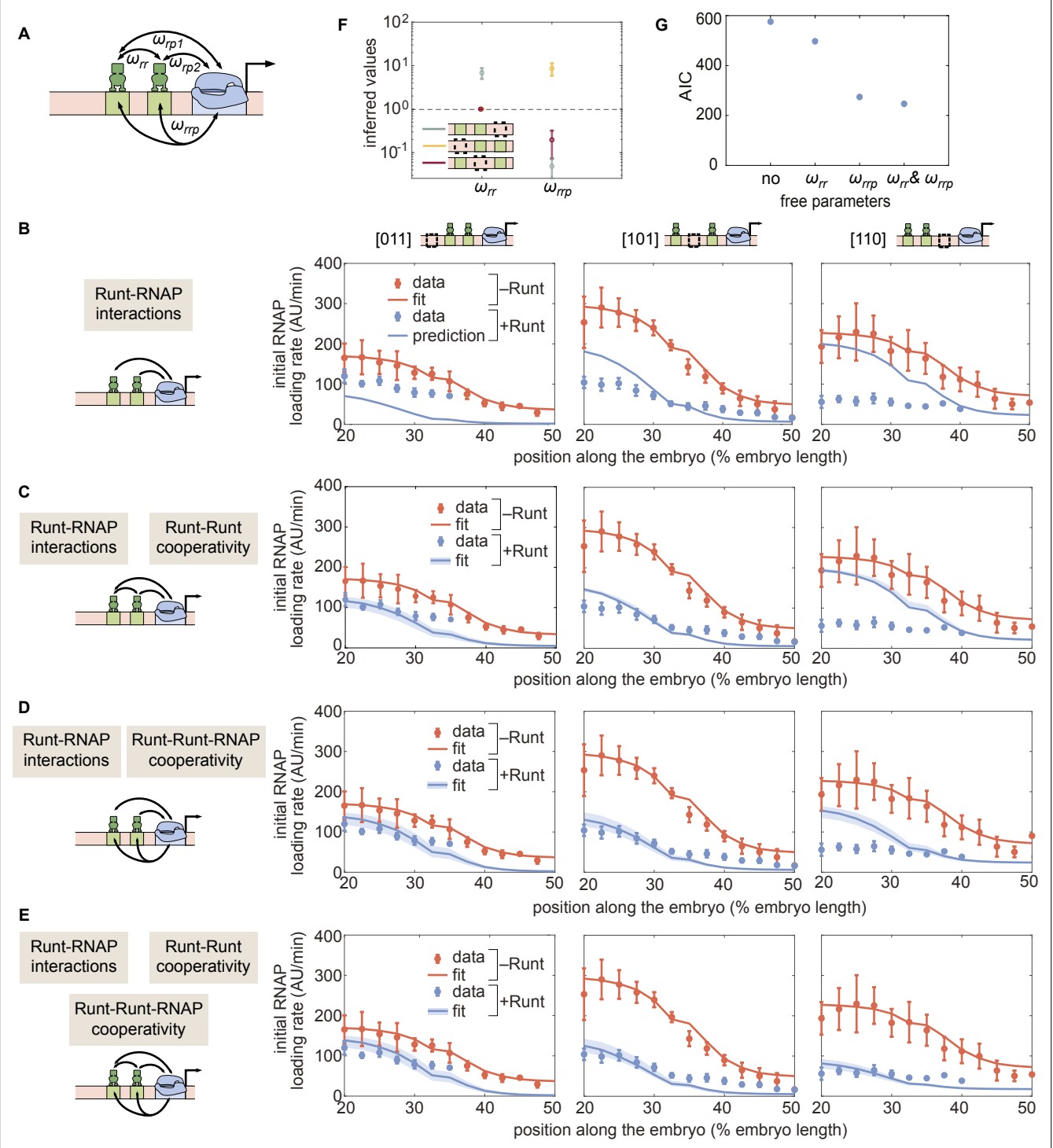

**Figure 6.** Prediction for the transcription initiation rate of *hunchback* P2 with two-Runt binding sites under different models of cooperativity. (**A**) Direct repression model for *hunchback* P2 with two Runt binding sites featuring Runt-RNAP interaction terms given by $\omega_{rp1}$ and, $\omega_{rp2}$ Runt-Runt cooperativity captured by $\omega_{rr}$, and Runt-Runt-RNAP higher-order cooperativity accounted for by $\omega_{rrp}$. (**B**) Parameter-free model prediction for two Runt binding sites when the two Runt molecules bind the DNA and interact with RNAP independently of each other. (**C,D,E**) Best MCMC fits for the data for two-Runt binding site constructs for models with various combinations of cooperativity parameters. (**C**) Model incorporating Runt-Runt cooperativity. (**D**) Model incorporating Runt-Runt-RNAP higher-order cooperativity. (**E**) Model accounting for both Runt-Runt cooperativity and Runt-Runt-RNAP higher-order cooperativity. (**F**) Fixed or inferred parameters $\omega_{rr}$ and $\omega_{rrp}$ for all two-Runt binding site constructs. Note that $\omega_{rr}$ is fixed to 1 for [011] and [101] constructs due to the fact that no Runt-Runt cooperativity is necessary to quantitatively describe the expression driven by these constructs; only the [110]

*Figure 6 continued on next page*

*Figure 6 continued*

construct is used to infer both $\omega_{rr}$ and $\omega_{rrp}$. The horizontal line of $\omega = 1$ denotes the case of no cooperativity. (**G**) Akaike Information Criterion (AIC) for all four scenarios of different free parameters shown throughout (**B–E**). (B-E, data points represent mean and standard error of the mean over the embryos. C-E, shaded error bars represent 95% confidence intervals for the best MCMC fits for the Runt WT datasets; F, data and error bars represent the mean and standard deviation of the posterior chain, while the standard deviation for the fixed $\omega_{rr}$ is set to 0.).

The online version of this article includes the following figure supplement(s) for figure 6:

**Figure supplement 1.** Prediction for two-Runt binding sites constructs based on the inferred parameters from the one-Runt binding site cases for different modes of repression for the (**A**) [011], (**B**) [101], and (**C**) [110] constructs.

**Figure supplement 2.** Prediction for *hunchback* P2 transcription initiation rate with two-Runt binding sites under the competition scenario for different combinations of cooperativities.

**Figure supplement 3.** Prediction for *hunchback* P2 transcription initiation rate with two-Runt binding sites under the quenching mechanism for different combinations of cooperativities.

**Figure supplement 4.** Invoking Runt-Runt cooperativity in the thermodynamic model is not sufficient to explain the experimental data from *hunchback* P2 with two Runt binding sites.

**Figure supplement 5.** Statistical mechanics model incorporating Runt-Runt-RNAP higher-order cooperativity.

**Figure supplement 6.** Invoking Runt-Runt cooperativity and higher-order cooperativity can explain the experimental data from *hunchback* P2 with two Runt binding sites.

**Figure supplement 7.** Sensitivity test for $K_r$ by repeating the MCMC inference for different scenarios of cooperativities with different values of $K_r$.

$$
\begin{aligned}
Rate \;=\; & R\left(p + b^6 p\omega_{bp} + rp(\omega_{rp1} + \omega_{rp2}) + r^2 p\omega_{rp1}\omega_{rp2}\omega_{rr}\omega_{rrp} + b^6 rp\omega_{bp}(\omega_{rp1} + \omega_{rp2})+ \right. \\
& b^6 r^2 p\omega_{bp}\omega_{rp1}\omega_{rp2}\omega_{rr}\omega_{rrp}\Big)\left(1 + b^6(1 + 2r + p\omega_{bp}) + 2r + p + rp(\omega_{rp1} + \omega_{rp2}) + r^2(\omega_{rr}\right. \\
& \left. + p\omega_{rp1}\omega_{rp2}\omega_{rr}\omega_{rrp}) + b^6 rp\omega_{bp}(\omega_{rp1} + \omega_{rp2}) + b^6 r^2\omega_{rr} + b^6 r^2 p\omega_{bp}\omega_{rp1}\omega_{rp2}\omega_{rr}\omega_{rrp}\right)^{-1}.
\end{aligned}
\tag{6}
$$

Here, once again, we have color-coded parameters to be inferred in red to differentiate them from fixed parameters that were already inferred in previous sections. Despite the complexity of this equation, note that its only free parameters are the cooperativity parameters $\omega_{rr}$ and $\omega_{rrp}$. As a result, we sought to determine whether the Runt-RNAP cooperativity terms, $\omega_{rp1}$ and $\omega_{rp2}$, are sufficient to predict repression by two Runt molecules, or whether the cooperativities given by $\omega_{rr}$ and $\omega_{rrp}$ also need to be invoked.

Consider the simplest case where two Runt molecules bind and interact with RNAP independently from each other. Here, $\omega_{rr} = 1$, and $\omega_{rrp} = 1$. This model has no free parameters; all parameters have already been determined by the inferences performed on Runt null datasets and one-Runt binding site constructs (*Figure 4* and *Figure 5*, respectively). While there was some agreement between the model and the data for the [101] construct (*Figure 6B*, center), significant deviations from the prediction occurred for the other two constructs. These deviations ranged from less repression than predicted for [011] (*Figure 6B*, left) to more repression than predicted for [110] (*Figure 6B*, right). Thus, this simple model of Runt independent repression is not supported by the experimental data, suggesting additional regulatory interactions between the Runt molecules and RNAP.

A first alternative to the independent repression model is the consideration of Runt-Runt cooperative interactions such as those that characterize many transcription factors (*Park et al., 2019*; *Estrada et al., 2016b*; *He et al., 2010*; *Segal et al., 2008*; *Ptashne, 2004*). However, adding a Runt-Runt cooperativity term, $\omega_{rr}$, was insufficient to account for the observed regulatory behavior (*Figure 6C*; *Figure 6—figure supplement 4* more thoroughly analyzes this discrepancy). A second alternative consists in incorporating a Runt-Runt-RNAP higher-order cooperativity term, $\omega_{rrp}$. While the best MCMC fits revealed significant improvements in predictive power, important deviations still existed for the [110] construct (*Figure 6D*, right; *Figure 6—figure supplement 5* more thoroughly analyzes the MCMC inference results).

Not surprisingly, given the agreement of the higher-order cooperativity model with the data for the [011] and [101] constructs (*Figure 6D*, left and center), this agreement persisted when both Runt-Runt cooperativity and Runt-Runt-RNAP higher-order cooperativity were considered (*Figure 6E*, left and center). However, including these two cooperativities also significantly improved the ability of the model at explaining the [110] experimental data (*Figure 6E*, right). Thus, while higher-order cooperativity is the main interaction necessary to quantitatively describe repression by two Runt repressors,

pairwise cooperativity also needs to be invoked. This conclusion is supported by our MCMC sampling: posterior distributions for the Runt-Runt cooperativity term are not well constrained for the [011] or [101] constructs, whereas Runt-Runt-RNAP higher-order cooperativity is constrained very well across all constructs (*Figure 6—figure supplement 6D*; *Figure 6—figure supplement 6* more thoroughly analyzes the MCMC inference results). As a result, accounting for both pairwise and higher-order cooperativity is necessary for the model to explain the observed rate of RNAP loading of all three constructs.

The higher-order cooperativity revealed by our analysis can lead to more or less repression than predicted by the independent repression model, motivating us to determine the magnitude of this cooperativity across constructs. To make this possible, we inferred the magnitude of the Runt-Runt cooperativity $\omega_{rr}$ and the Runt-Runt-RNAP higher-order cooperativity $\omega_{rrp}$. As shown in *Figure 6F*, depending on the spatial arrangement of Runt binding sites, the Runt-Runt-RNAP higher-order cooperativity term $\omega_{rrp}$ can be below or above 1. Note that, in doing these fits, we first set the Runt-Runt cooperativity, $\omega_{rr}$, values for [011] and [101] to 1 because, as we had demonstrated in *Figure 6D*, only the higher-order Runt-Runt-RNAP cooperativity was necessary. Thus, different placements of Runt molecules on the enhancer lead to distinct higher-order interactions with RNAP which, in turn, can result in less or more repression than predicted by a model where Runt molecules act independently of each other.

## Repression by three-Runt binding sites also requires higher-order cooperativity

Building on our success in deploying thermodynamic models to explain repression by one- and two-Runt binding sites, we investigated repression by three-Runt binding sites. First, we accounted for pairwise interactions between Runt and RNAP, which were inferred from measurements of the one-Runt binding site constructs (*Figure 1B*), yielding $\omega_{rp_{[001]}}$, $\omega_{rp_{[010]}}$, and $\omega_{rp_{[100]}}$ from [001], [010], and [100]. Second, we considered pairwise protein-protein interactions between Runt molecules (*Figure 1C*), which were inferred from the two-Runt binding sites constructs through the parameters $\omega_{rr_{[011]}}$, $\omega_{rr_{[101]}}$, and $\omega_{rr_{[110]}}$. Finally, we incorporated Runt-Runt-RNAP higher-order cooperativity acquired from the

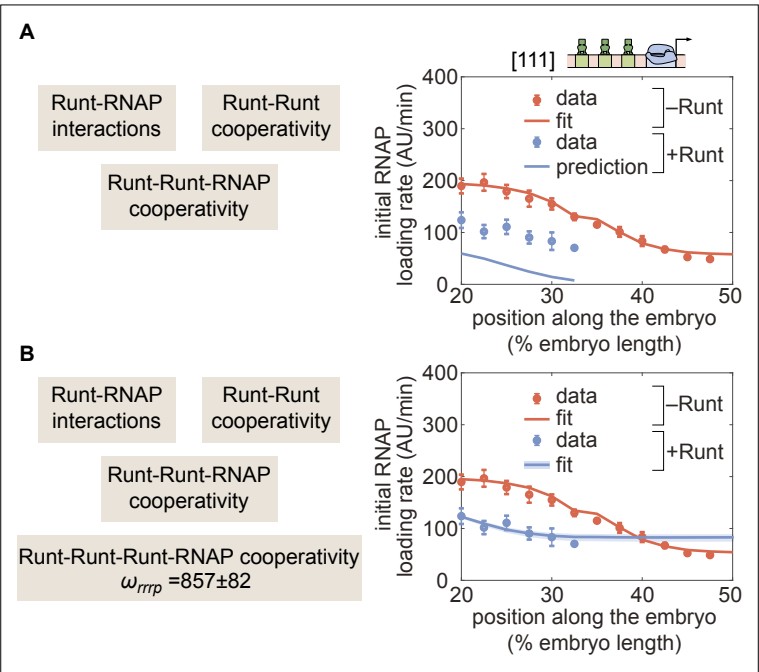

**Figure 7.** Prediction for *hunchback* P2 with three-Runt binding sites and multiple sources of cooperativity. (**A**) Prediction using previously inferred Runt-RNAP, Runt-Runt, and Runt-Runt-RNAP cooperativity parameters. (**B**) Best MCMC fit obtained by incorporating an additional Runt-Runt-Runt-RNAP higher-order cooperativity parameter of $\omega_{rrrp} = 857$, corresponding to roughly 7 $k_B T$ of free energy. (A,B, data points represent mean and standard error of the mean over >3 embryos; B, shaded regions represent 95% confidence intervals for the best MCMC fit.).

two-Runt binding sites constructs (*Figure 1C*) captured by $\omega_{rrp_{[011]}}$, $\omega_{rrp_{[101]}}$, and $\omega_{rrp_{[110]}}$. we tested our model predictions using a similar scheme to that described in the previous section: we generated a parameter-free prediction for the initial rate of transcription by using the inferred parameters from the one- and two-Runt binding sites constructs, including the pairwise and higher-order interactions described above.

*Figure 7A* shows the resulting parameter-free prediction. As seen in the figure, our model could not qualitatively recapitulate the experimental data as it predicted too much repression. Such disagreement suggests that additional regulatory interactions are at play. Building on the need for higher-order cooperativity in the two-Runt binding site case, we propose the existence of higher-order cooperativities necessary to describe regulation by three Runt molecules—Runt-Runt-Runt higher-order cooperativity, $\omega_{rrr}$ and Runt-Runt-Runt-RNAP higher-order cooperativity, $\omega_{rrrp}$ (*Figure 1D*). The resulting expression for the predicted rate of transcription in the presence of all these sources of cooperativity is shown in *Equation S10* in Section 'Derivation of the general and simpler thermodynamic model for the hunchback P2 enhancer with one Runt binding site'. For simplicity, we assumed that the Runt-Runt-Runt cooperativity is one, and only determined the Runt-Runt-Runt-RNAP higher-order cooperativity. By including only a Runt-Runt-Runt-RNAP higher-order cooperativity parameter, our model recapitulated the experimental data (*Figure 7B*). Thus, our results further support the view in which the addition of Runt repressor binding motifs in an enhancer calls for the incorporation of cooperativities of increasingly higher-order.

## Discussion

One of the challenges in generating predictions to probe thermodynamic models is that, often, these models are contrasted against experimental data from endogenous regulatory regions (*Segal et al., 2008*; *Sayal et al., 2016*; *Park et al., 2019*). Here, the presence of multiple binding sites for several transcription factors—known and unknown (*Vincent et al., 2016*)—leads to models with a combinatorial explosion of free parameters. Like the proverbial elephant that can be fit with four parameters (*Mayer et al., 2010*), experiments with endogenous enhancers typically contain enough parameters to render it possible to explain away apparent disagreement between theory and experiment (*Garcia et al., 2020*).

To close this gap, synthetic minimal enhancers have emerged as an attractive alternative to endogenous enhancers (*Fakhouri et al., 2010*; *Sayal et al., 2016*; *Park et al., 2019*; *Crocker et al., 2016*). Here, the presence of only a handful of transcription factor binding sites and the ability to systematically control their placement and affinity dramatically reduce the number of free parameters in the model (*Garcia et al., 2020*). Inferences performed on these synthetic constructs could then inform model parameters that would make it possible to quantitatively predict transcriptional output of de novo enhancers (*Sayal et al., 2016*).

Building on these works, we sought to predict how the Runt repressor, which counteracts activation by Bicoid along the anterior-posterior axis of the early fly embryo (*Chen et al., 2012*), dictates output levels of transcription. To dissect repression, a strong and detectable level of expression in the absence of the repressor was needed, prompting us to choose a simple system of synthetic enhancers based on the strong *hunchback* P2 minimal enhancer (*Garcia et al., 2013*; *Chen et al., 2012*). This enhancer has been carefully studied in terms of its activator Bicoid and the pioneer-like transcription factor Zelda in the early embryo (*Driever and Nüsslein-Volhard, 1988*; *Garcia et al., 2013*; *Park et al., 2019*; *Eck et al., 2020*), making it easier to identify neutral sequences within the enhancer for introducing Runt binding sites (*Chen et al., 2012*). Further, when inserted into *hunchback* P2, Runt binding site number determines the level of transcription incrementally (*Chen et al., 2012*). Thus, *hunchback* P2 provided an ideal scaffold for quantitatively and systematically dissecting repression by Runt.

Previous studies using synthetic enhancers relied on measurements of input transcription factor patterns using fluorescence immunostaining, and of cytoplasmic mRNA patterns using fluorescence in situ hybridization (FISH) or single-molecule FISH. These fixed-tissue techniques have key differences from the live-imaging approach adopted here. First, given the dynamical nature of development, it is necessary to know when data were acquired. Doing so with high temporal resolution using FISH is challenging, although it can be accomplished to some degree by synchronizing embryo deposition before fixation (*Park et al., 2019*). Second, while most transcription factors directly dictate the

rate of RNAP loading, and hence the rate of mRNA production (*Spitz and Furlong, 2012*; *Garcia et al., 2013*; *Eck et al., 2020*), typical FISH measurements report on the accumulated mRNA in the cytoplasm, which is a convolution of all processes of the transcription cycle—initiation, elongation, and termination (*Liu et al., 2021*; *Alberts et al., 2015*)—as well as mRNA nuclear export dynamics, diffusion, and degradation. These processes could be modulated in space and time, potentially confounding measurements. Here, we overcame these challenges by using the MS2 technique to precisely time our embryos and acquire the rate of transcription initiation. Of course, despite the ease of measuring the rate of transcription initiation using MS2, the accumulated mRNA is presumably a more relevant quantity for predicting downstream cellular decision making. Previous studies have shown that the MS2-MCP technique can also be used to quantify such patterns of accumulated mRNA, and that this quantification leads to results comparable to those obtained by smFISH (*Garcia et al., 2013*; *Lammers et al., 2020*). Following the same quantification method, we assessed the relationship between the initial rate of RNAP loading and the accumulated mRNA (*Figure 3—figure supplement 5*, *Figure 3—figure supplement 6*) by plotting them against each other. Reassuringly, as shown in *Figure 3—figure supplement 8*, our analysis revealed a strong correlation (with Pearson's correlation coefficient of 0.90), supporting our claim that higher-order cooperativity is essential for explaining the action of multiple transcription factors during the development.

Interestingly, our initial dissection of constructs containing various combinations of Runt binding sites, but in the absence of Runt protein, revealed that unrepressed gene expression levels depend strongly on the number and placement of the binding sites within the enhancer (*Figure 4A*). These results challenge previous assumptions that unregulated gene expression levels stay unchanged as enhancer architecture is modulated (*Sayal et al., 2016*; *Fakhouri et al., 2010*; *Barr et al., 2017*), but they are in accordance with observations in bacterial systems (*Garcia et al., 2012*). As a result, our measurements call for accounting for unregulated levels in future quantitative dissections of eukaryotic enhancers, or to study relative magnitudes such as the fold-change in gene expression that has driven the dissection of bacterial transcriptional regulation (*Phillips et al., 2019*).

Using the thermodynamic model shown in *Equation 3*, we determined that the Bicoid-dependent parameters remain constant while RNAP-dependent parameters vary across these synthetic enhancer constructs. We speculate that the overall enhancer sequence, which changed as a result of the placement of different combinations of Runt binding sites within it, might affect the binding of the transcriptional machinery. Specifically, since the enhancer is proximal to the promoter, the transcriptional machinery might see slightly different DNA sequences in the vicinity of the promoter as suggested by published structures of the transcriptional machinery assembled on DNA (*Louder et al., 2016*).

**Table 1.** Interaction energies for the Runt-related cooperativity parameters from one-, two-, and three-Runt sites constructs.

Note that we used the Boltzmann relation of $\omega = exp(-E/(k_B T))$, where the $E$ is the interaction energy, $k_B$ is the Boltzmann constant, and $T$ is the temperature.

**Interaction energies for the Runt-related cooperativity parameters**

| model parameter | construct | interaction energy ($K_B T$) |
|---|---|---|
| | [001] | 2.34 ± 0.63 |
| | [010] | 1.36 ± 0.36 |
| Runt-RNAP interaction,$\omega_{rp}$ | [100] | 0.18 ± 0.24 |
| | [011] | 0 (manually set) |
| | [110] | -0.95 ± 0.12 |
| Runt-Runt interaction,$\omega_{rr}$ | [101] | 0 (manually set) |
| | [011] | -2.09 ± 0.27 |
| | [110] | 4.15 ± 1.14 |
| Runt-Runt-RNAP interaction,$\omega_{rrp}$ | [101] | 1.12 ± 0.51 |
| Runt-Runt-Runt-RNAP interaction,$\omega_{rrrp}$ | [111] | -2.12 ± 0.14 |

Once we accounted for this difference in unrepressed gene expression levels, we determined that the repression profiles obtained for constructs bearing one-Runt binding site could be described by a simple thermodynamic model (*Figure 2*). Specifically, we showed that the same dissociation constant described Runt binding regardless of the position of its binding site along the enhancer (*Figure 5A*). Further, the Runt-RNAP interaction terms describing repressor action decreased as the binding site was placed farther from the promoter (*Figure 5C*), qualitatively consistent with a 'direct repression' model in which Runt needs to physically contact RNAP in order to realize its function (*Jaynes and O'Farrell, 1991*; *Gray et al., 1994*; *Hewitt et al., 1999*).

Although our model recapitulated repression by a one-Runt binding site, the inferred parameters were insufficient to quantitatively predict repression by two-Runt binding sites (*Figure 6B*). These results suggest that multiple repressors do not act independently of each other. Instead, new parameters describing both Runt-Runt cooperativity and Runt-Runt-RNAP higher-order cooperativity had to be incorporated into our models to quantitatively describe Runt action in these constructs (*Figure 6— figure supplement 1C–E*). An examination of the various cooperativity values inferred in the language of interaction energies (*Table 1*) revealed that these energies were of a magnitude comparable to protein-protein interaction energies previously measured in bacterial systems (*Dodd et al., 2004*; *Bintu et al., 2005a*; *Amit et al., 2011*). Interestingly, these interaction energies were both positive and negative, suggesting that both cooperativity or anti-cooperativity are at play depending on enhancer architecture (*Amit et al., 2011*). Additionally, the [101] construct showed a closer agreement with the parameter-free prediction, without invoking higher-order cooperativity, than the other two constructs ([110] or [011]). This further supports a picture where higher-order cooperativity is sensitive to the placement and orientation of transcription factor binding sites within regulatory regions.

While we have long known about protein-protein cooperative interactions (*Ackers et al., 1982*), in the last few years it has become clear that higher-order cooperativity can also be at play in eukaryotic systems (*Estrada et al., 2016a*; *Park et al., 2019*; *Biddle et al., 2020*) as well as in bacteria (*Dodd et al., 2004*) and archaea (*Peeters et al., 2013*). The existence of this higher-order cooperativity suggests that, to predict gene expression from DNA sequence, it might be necessary to build an understanding of the many simultaneous interactions that precede transcriptional initiation. Our discovery of higher-order cooperativity in the action of multiple Runt molecules opens up new avenues to uncover the molecular nature of this phenomenon. For example, following an approach developed in *Park et al., 2019*, it could be possible to determine whether and how these cooperativity parameters are modulated upon perturbation of molecular players such as the Groucho or CtBP co-repressors, Big-brother, a co-factor facilitating the Runt binding to DNA, and components of the mediator complex (*Park et al., 2019*; *Courey and Jia, 2001*; *Walrad et al., 2011*). Indeed, *Park et al., 2019* recently showed that co-activators and mediator units are involved in dictating the magnitude of similar higher-order cooperativity terms in activation by Bicoid. Thus, our thermodynamic models provide a lens through which to dissect the molecular underpinnings of Runt interactions with itself and with the transcriptional machinery.

Notably, the need to invoke cooperative interactions as more Runt binding sites are being added opposes our goal of predicting complex regulatory architectures from experiments with simpler architectures without the need to invoke new parameters. However, it will be interesting to determine whether more parameters need to be invoked as the number of Runt binding sites increases beyond three, or whether the parameters already inferred are sufficient to endow our models with parameter-free predictive power.

Importantly, while our model adopted a 'direct repression' view of the mechanism of Runt action, other mechanisms of repression such as 'quenching' could also describe the data. While all such models call for higher-order cooperativity to describe the data (Supplementary Section 'Comparison of different modes of repression'), our data cannot differentiate among those models. Thus, we did not attempt to distinguish different molecular mechanisms of Runt transcriptional repression.

Finally, even though the work presented here has relied exclusively on thermodynamic models, it is important to note that a much more general approach based on kinetic models that are not in thermodynamic equilibrium could also be appropriate for describing our data. Indeed, an increasing body of work over the last few years has provided evidence for the necessity of invoking these more complex models in the context of transcriptional regulation in eukaryotes (*Estrada et al., 2016a*; *Li et al., 2018*; *Park et al., 2019*; *Eck et al., 2020*). In future work, it will be interesting to determine

whether, when our data is viewed through the lens of these non-equilibrium models, invoking higher-order cooperativity is still necessary or whether, instead, simple pairwise protein-protein interactions suffice to reach an agreement between theory and experiment.

Overall, the work presented here establishes a framework for systematically and quantitatively studying repression in the early fly embryo. As showcased here, synthetic enhancers based on the *hunchback* P2 minimal enhancer constitute an ideal scaffold for the study of other repressors in early fly embryos. For example, we envision that this approach could be used to dissect repression by other transcription factors such as *Capicua* or *Krüppel* (*Löhr et al., 2009*; *Sauer and Jäckle, 1991*; *Papagianni et al., 2018*; *Chen et al., 2012*), and to probe observations of multiple repressors working together to oppose activation by Bicoid in establishing gene expression patterns along the anterior-posterior axis (*Chen et al., 2012*; *Briscoe and Small, 2015*). We anticipate that a similar approach could be used to dissect repression along the dorso-ventral axis of the embryo, by for example, adding repressor binding sites to well-established reporter constructs that are only regulated by the Dorsal activator (*Jiang and Levine, 1993*). Critically, we need to understand not only how one species of repressor works in concert with an activator, but also how multiple species of repressors work together as a system. The approach presented here provides a way forward for predictively understanding the complex gene regulatory network that shapes gene expression patterns in the early fly embryo.

## Materials and methods
### Generation of synthetic enhancers with MS2 reporter

The synthetic enhancer constructs used in this study are based off of *Chen et al., 2012*. In summary, the *hunchback* P2 enhancer was used as a scaffold to introduce Runt binding sites at different positions that are thought to be neutral (i.e. these Runt binding sites do not interfere with any other obvious binding sites for other transcription factors in the early *Drosophila* embryos as shown in *Figure 4—figure supplement 1*). For the three positions chosen to introduce Runt binding sites in *Chen et al., 2012*, the Gene Synthesis service from Genscript was used to generate synthetic enhancers with all possible configurations of zero-, one-, two-, and three-Runt binding sites in *hunchback* P2 as shown in *Figure 1A*. The enhancer sequences were placed into the original plasmid pIB backbone (*Chen et al., 2012*) using the Gene Fragment Synthesis service in Genscript, followed by the *even-skipped* promoter, and 24 repeats of the MS2v5 loop (*Wu et al., 2015*), the *lacZ* coding sequence, and the $\alpha$-Tubulin 3'UTR sequence (*Chen et al., 2012*) as shown in *Table 2*. These plasmids were injected into the 38F1 landing site using the RMCE method (*Bateman et al., 2006*) by BestGene Inc Flies

**Table 2.** List of plasmids used to create the transgenic fly lines used in this study.

**Plasmids**

| Name (hyperlinked to Benchling) | Function |
| --- | --- |
| pIB-hbP2-evePr-MS2v5-LacZ-Tub3UTR | [000]-MS2v5 reporter construct |
| pIB-hbP2+r1-far-evePr-MS2v5-LacZ-Tub3UTR | [100]-MS2v5 reporter construct |
| pIB-hbP2+r1-mid-evePr-MS2v5-LacZ-Tub3UTR | [010]-MS2v5 reporter construct |
| pIB-hbP2+r1-close-evePr-MS2v5-LacZ-Tub3UTR | [001]-MS2v5 reporter construct |
| pIB-hbP2+r2-2+3-evePr-MS2v5-LacZ-Tub3UTR | [011]-MS2v5 reporter construct |
| pIB-hbP2+r2-1+3-evePr-MS2v5-LacZ-Tub3UTR | [101]-MS2v5 reporter construct |
| pIB-hbP2+r2-1+2-evePr-MS2v5-LacZ-Tub3UTR | [110]-MS2v5 reporter construct |
| pIB-hbP2+r3-evePr-MS2v5-LacZ-Tub3UTR | [111]-MS2v5 reporter construct |
| pHD-scarless-LlamaTag-Runt | Donor plasmid for LlamaTag-Runt CRISPR knock-in fusion for the N-terminal |
| pU6:3-gRNA(Runt-N-2) | gRNA plasmid for LlamaTag-Runt CRISPR knock-in fusion for the N-terminal |
| pCasper-vasa-eGFP | *vasa* maternal driver for ubiquitous eGFP expression in the early embryo |

were screened by selecting for white eye color and made homozygous. The orientation of the insertion was determined by genomic PCR to ensure a consistent orientation across all of our constructs. Specifically, we used two sets of primers that each amplified one of these two possible orientations: 'Upward', where the forward primer binds to a genomic location outside of 38F1 (TTCTAGTTCCAGTGAAATCCAAGCA) and the reverse primer binds to a location in our reporter transgene (ACGCCAGGGTTTTCCCAG), and 'Downward', where the forward primer remains the same as the 'Upward' set and the reverse primer binds to a location in our reporter transgene (CTCTGTTCTCGCTATTATTCCAACC) when the insertion is the opposite orientation to the 'Upward' orientation. As a result, only amplicons from either one of the orientations of insertion in the 38F1 landing site can be obtained. We chose the 'Downward' orientation for all our constructs.

## CRISPR-Cas9 knock-in of the green LlamaTag in the endogenous *runt* locus

We used CRISPR-Cas9 mediated Homology Directed Repair (HDR) to insert the LlamaTag against eGFP into the N-terminal of the *runt* endogenous locus (*Bothma et al., 2018*; *Gratz et al., 2015*). The donor plasmid was constructed by stitching individual fragments—PCR amplified left/right homology arms from the endogenous *runt* locus roughly 1 kb in length each, LlamaTag, and pHD-scarless vector—using Gibson assembly (*Gratz et al., 2015*). The PAM sites in the donor plasmid were mutated such that the Cas9 only cleaved the endogenous locus, not the donor plasmid, without changing the amino acid sequence of the Runt protein. The final donor plasmid contained the 3xP3-dsRed marker such that dsRed is expressed in the fly eye and ocelli for screening. Positive transformant flies were screened using a fluorescence dissection scope and set up for single fly crosses to establish individual lines that were then verified with PCR amplification and Sanger sequencing (UC Berkeley Sequencing Facility). Importantly, this *llamaTag-runt* allele rescues development to adulthood as a homozygous. Thus we concluded that the LlamaTag-Runt allele can be used to monitor the behavior of endogenous Runt protein.

## Fly strains

Transcription from the synthetic enhancer reporter constructs was measured by using embryos from crossing *yw;his2av-mRFP1;MCP-eGFP(2)* females and *yw;synthetic enhancer-MS2v5-lacZ;+* males as described in *Garcia et al., 2013*; *Eck et al., 2020*; *Lammers et al., 2020*.

eGFP-Bicoid measurements were performed using the fly line from *Gregor et al., 2007*. The LlamaTag-Runt measurements were done using the fly line *LlamaTag-Runt; +; vasa-eGFP, His2Av-iRFP*

**Table 3.** List of fly lines used in this study and their experimental usage.

**Fly lines**

| Genotype | Use |
| --- | --- |
| LlamaTag-Runt; +; vasa-eGFP, His2Av-iRFP | Visualize LlamaTagged Runt protein and label nuclei |
| LlamaTag-Runt; +; MCP-eGFP(4F), His2Av-iRFP | Visualize LlamaTagged Runt protein, nascent transcripts and label nuclei |
| run3/FM6; +; + | Visualize LlamaTagged Runt protein, nascent transcripts and label nuclei |
| yw; His2Av-mRFP; MCP-eGFP | Females to label nascent RNA and nuclei |
| yw; [000]-MS2v5; + | Males carrying the MS2 reporter transgene |
| yw; [100]-MS2v5; + | Males carrying the MS2 reporter transgene |
| yw; [010]-MS2v5; + | Males carrying the MS2 reporter transgene |
| yw; [001]-MS2v5; + | Males carrying the MS2 reporter transgene |
| yw; [011]-MS2v5; + | Males carrying the MS2 reporter transgene |
| yw; [101]-MS2v5; + | Males carrying the MS2 reporter transgene |
| yw; [110]-MS2v5; + | Males carrying the MS2 reporter transgene |
| yw; [111]-MS2v5; + | Males carrying the MS2 reporter transgene |

illustrated in *Table 3*. Briefly, eGFP was supplied by a *vasa* maternal driver. Females carrying both the LlamaTag-Runt and the *vasa*-driven eGFP were crossed with males carrying the LlamaTag-Runt, the progeny from this cross were imaged and then recovered to determine the embryo's sex using PCR. PCR was run with three sets of primers: Y chr1 (Forward: CGATCCAGCCCAATCTCTCATATCACTA, Reverse: ATCGTCGGTAATGTGTCCTCCGTAATTT), Y chr2 (Forward: AACGTAACCTAGTCGGATTG CAAATGGT, Reverse: GAGGCGTACAATTTCCTTTCTCATGTCA), and Auto1 (Forward: GATTCGAT GCACACTCACATTCTTCTCC, Reverse: GCTCAGCGCGAAACTAACATGAAAAACT). Two of primers in the set (Y chr1 and Y chr2) bind to the Y chromosome while the other one (Auto1) binds to one of the autosomes and constitutes a positive control (*Lott et al., 2011*). The Histone-iRFP fly line was from *Pan et al., 2022*, and was used for nuclei segmentation to extract nuclear flourescence from the eGFP channel.

To generate the embryos that are zygotic null for the *runt* allele, we used a fly cross scheme consisting of two crosses. In the first generation, we crossed *LlamaTag-Runt;+;+* males with *run3/ FM6;+;MCP-eGFP(4 F),his2av-mRFP1* females. *run3* is the null allele for *runt*, missing around 5 kb including the coding sequence of the *runt* locus (*Gergen and Butler, 1988*; *Chen et al., 2012*). The *MCP-eGFP(4 F)* transgene expresses approximately twice the amount of MCP protein than the *MCP-eGFP(2)* (*Garcia et al., 2013*; *Eck et al., 2020*) and thus results in similar levels of MCP to those of *MCP-eGFP(2)* in the trans-heterozygotes. The female progeny from this cross, *LlamaTag-Runt/ run3;+;MCP-eGFP(4 F),his2av-mRFP1/+* was then crossed with males whose genotype was *LlamaTag-Runt/Y;synthetic enhancer-MS2v5-lacZ;+* to produce the embryos that we used for live imaging. The resulting embryos carried maternally supplied MCP-eGFP and His-RFP for visualization of nascent transcripts and nuclei. The X chromosome contained a LlamaTag-Runt allele or *run3* null allele. We could differentiate between these two genotypes because, when the embryo had the Runt allele, a stripe pattern would appear in late nc14. We imaged all embryos until late nc14 to make sure that we were capturing the nulls.

## Sample preparation and data collection

Sample preparation was done following the protocols described in *Garcia et al., 2013*. Briefly, embryos were collected, dechorionated with bleach for 1–2 min, and then mounted between a semiperme-able membrane (Lumox film, Starstedt, Germany) and a coverslip while embedded in Halocarbon 27 oil (Sigma-Aldrich). Live imaging was performed using a Leica SP8 scanning confocal microscope, a White Light Laser and HyD dectectors (Leica Microsystems, Biberach, Germany). Imaging settings for the MS2 experiments with the presence of MCP-eGFP and Histone-RFP were the same as in *Eck et al., 2020* except that we used a 1024x245 pixel format to image a wider field of view along the anterior-posterior axis. The settings for the eGFP-Bicoid measurements were the same as described in *Eck et al., 2020*.

The settings for the eGFP:LlamaTag-Runt measurements were similar to that of eGFP-Bicoid except for the following. To increase our imaging throughput, we utilized the 'Mark and Position' functionality in the LASX software (Leica SP8) to image 5–6 embryos simultaneously. To account for the decreased time resolution, we lowered the z-stack size from 10 μm to 2.5 μm, keeping the 0.5 μm z-step. By doing this, we could maintain 1-min frame rate for each imaged embryo. Additionally, these flies expressed Histone-iRFP, instead of Histone-RFP as in *Eck et al., 2020*, so that we used a 670 nm laser at 40 μW (measured at a 10x objective) for excitation of the histone channel, and the HyD detector was set to a 680 nm-800 nm spectral window (*Figure 3—figure supplement 7*).

## Image analysis

Images were analyzed using custom-written software (MATLAB, mRNA Dynamics Github repository; *Garcia Lab @ UC Berkeley, 2022*) following the protocol in *Garcia et al., 2013* and *Eck et al., 2020*. Briefly, this procedure involved segmentation and tracking of nuclei and transcription spots. First, segmentation and tracking of individual nuclei were done using the histone channel as a nuclear mask. Second, segmentation of each transcription spot was done based on its fluorescence intensity and existence over multiple z-stacks. The intensity of each MCP-GFP transcriptional spot was calculated by integrating pixel intensity values in a small window around the spot and subtracting the background fluorescence measured outside of the active transcriptional locus. When there was no detectable tran-scriptional activity, we assigned NaN values for the intensity. The tracking of transcriptional spots was

done by using the nuclear tracking and proximity of transcriptional spots between consecutive time points. The nuclear protein fluorescence intensities from the eGFP-Bicoid and LlamaTag-Runt fly lines, which we use as a proxy for the protein nuclear concentration, were calculated as follows. Using the nuclear mask generated from the histone channel, we performed the same nuclear segmentation and tracking as described above for the MS2 spots. Then, for every z-section, we extracted the integrated fluorescence over a $2\mu m$ diameter circle on the xy-plane centered on each nucleus. For each nucleus, the recorded fluorescence corresponded to the z-position where the fluorescence was maximal. This resulted in an average nuclear concentration as a function of time for each single nucleus. These concentrations from individual nuclei were then averaged over a narrow spatial window (2.5% of the embryo length) to generate the spatially averaged protein concentration reported in the main text. For the eGFP:LlamaTag-Runt datasets, we had to subtract the background eGFP fluorescence due to the presence of an unbound eGFP population (*Bothma et al., 2018*). We used the same protocol described in *Bothma et al., 2018* and in the Supplementary Section 'Quantifying the nuclear concentration of LlamaTag-Runt' to extract this background.

### Bayesian inference procedure: Markov Chain Monte Carlo sampling

Parameter inference was done using the Markov Chain Monte Carlo (MCMC) method. We used a well-established package *MCMCstat* that uses an adaptive MCMC algorithm (*Haario et al., 2006*; *Haario et al., 2001*). A detailed description on how we performed the MCMC parameter inference, for example setting the priors and bounds for parameters, can be found in Supplementary Section 'Markov Chain Monte Carlo inference protocol'.

### Biological Materials

## Acknowledgements

We are grateful to Armando Reimer, Brandon Schlomann, Elizabeth Eck, Gabriella Martini, Jacques Bothma, Jeehae Park, Jia Ling, Matthew Norstad, Michael Eisen, Nicholas Lammers, Nipam Patel, Rob Phillips, Simon Alamos, Xavier Darzacq, and Yasemin Kirişçioğlu for their guidance and comments on our manuscript. This work was supported by the Burroughs Wellcome Fund Career Award at the Scientific Interface, the Sloan Research Foundation, the Human Frontiers Science Program, the Searle Scholars Program, the Shurl and Kay Curci Foundation, the Hellman Foundation, the NIH Director's New Innovator Award (DP2 OD024541-01), NSF CAREER Award (1652236), and an NIH R01 Award (R01GM139913) to HGG, and a KFAS scholarship to YJK. HGG is also a Chan Zuckerberg Biohub investigator.

## Additional information

### Funding

| Funder | Grant reference number | Author |
|---|---|---|
| Burroughs Wellcome Fund | Career Award | Hernan G Garcia |
| Sloan Research Foundation | | Hernan G Garcia |
| Human Frontier Science Program | | Hernan G Garcia |
| Searle Scholars Program | | Hernan G Garcia |
| Shurl and Kay Curci Foundation | | Hernan G Garcia |
| Hellman Foundation | | Hernan G Garcia |
| National Institute of Health | DP2 OD024541-01 | Hernan G Garcia |
| National Science Foundation | 1652236 | Hernan G Garcia |

| Funder | Grant reference number | Author |
|---|---|---|
| Korea Foundation for Advanced Studies | Graduate Student Fellowship | Yang Joon Kim |
| Chan Zuckerberg Biohub | Chan Zuckerberg Biohub Investigator Award | Hernan G Garcia |

The funders had no role in study design, data collection and interpretation, or the decision to submit the work for publication.

### Author contributions
Yang Joon Kim, Conceptualization, Resources, Data curation, Software, Formal analysis, Investigation, Visualization, Methodology, Writing – original draft, Writing – review and editing; Kaitlin Rhee, Data curation, Formal analysis, Investigation; Jonathan Liu, Software, Investigation, Writing – review and editing; Selene Jeammet, Investigation; Meghan A Turner, Resources; Stephen J Small, Conceptualization, Resources, Supervision; Hernan G Garcia, Conceptualization, Supervision, Funding acquisition, Investigation, Visualization, Writing – original draft, Project administration, Writing – review and editing

### Author ORCIDs
Yang Joon Kim http://orcid.org/0000-0003-1742-5657
Hernan G Garcia http://orcid.org/0000-0002-5212-3649

### Decision letter and Author response
Decision letter https://doi.org/10.7554/eLife.73395.sa1
Author response https://doi.org/10.7554/eLife.73395.sa2

## Additional files

### Supplementary files
• Transparent reporting form

### Data availability
All data (both input transcription factor concentration and output transcription from all synthetic enhancers, both pre- and post-processed data) have been deposited in Dryad under the DOI https://doi.org/10.5061/dryad.7sqv9s4sv.

The following dataset was generated:

| Author(s) | Year | Dataset title | Dataset URL | Database and Identifier |
|---|---|---|---|---|
| Kim Y, Rhee K, Liu J, Jeammet S, Turner M, Small S, Garcia HG | 2021 | Predictive modeling reveals that higher-order cooperativity drives transcriptional repression in a synthetic developmental enhancer | https://dx.doi.org/10.5061/dryad.7sqv9s4sv | Dryad Digital Repository, 10.5061/dryad.7sqv9s4sv |

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

## Appendix 1

### S1 Derivation of the general thermodynamic model for the *hunchback* P2 enhancer

In this section, we rederive the thermodynamic model presented in the main text, now without the assumption of strong Bicoid-Bicoid cooperativity. The equilibrium thermodynamic modeling framework that we used in this paper is described in more detail in *Bintu et al., 2005a*; *Bintu et al., 2005b*.

We start by modeling the case of *hunchback* P2 without any Runt binding sites, which is believed to have at least six Bicoid binding sites (*Park et al., 2019*; *Driever et al., 1989*). As shown by the states and weights presented in *Figure 2—figure supplement 1A*, in our thermodynamic model, we assume that the six Bicoid binding sites have the same dissociation constant given by $K_b$, and we posit that RNAP-promoter binding is governed by a dissociation constant given by $K_p$. We also assume pairwise cooperativity between Bicoid molecules given by $\omega_b$, and cooperativity between each Bicoid molecule and RNAP given by $\omega_{bp}$. For simplicity, we will use the dimensionless parameters $b = [Bicoid]/K_b$ and $p = [RNAP]/K_p$, where $[Bicoid]$, and $[RNAP]$ are the concentrations of Bicoid and RNAP, respectively, and $K_b$ and $K_p$ are their corresponding dissociation constants.

We factor the total partition function into two categories: $Z_b$ corresponding to states that only have Bicoid bound, and $Z_{bp}$ describing states with both Bicoid and RNAP bound. Then, we calculate each component separately. The sum of microstates for $Z_b$ is

$$Z_b = 1 + 6b + 15b^2\omega_b + \cdots + b^6\omega_b^5 = 1 + \sum_{i=1}^{6} \binom{6}{i}b^i\omega_b^{i-1}. \tag{S1}$$

Using the binomial theorem, we can simplify *Equation S1* leading to

$$Z_b = 1 + \sum_{i=1}^{6} \binom{6}{i}b^i\omega_b^{i-1} = 1 + \frac{1}{\omega_b}\left[(1 + b\,\omega_b)^6 - 1\right]. \tag{S2}$$

Using the same logic, we obtain $Z_{bp}$ such that

$$Z_{bp} = \left(p + p\sum_{i=1}^{6} \binom{6}{i}b^i\omega_b^{i-1}\omega_{bp}^i\right) = p + \frac{p}{\omega_b}\left[(1 + b\omega_b\omega_{bp})^6 - 1\right]. \tag{S3}$$

Using these two partition functions, we then calculate the probability of the promoter being bound by RNAP, $p_{bound}$ as

$$P_{bound} = \frac{Z_{bp}}{Z_b + Z_{bp}} = \frac{p + \frac{p}{\omega_b}\left[(1 + b\,\omega_b\,\omega_{bp})^6 - 1\right]}{1 + \frac{1}{\omega_b}\left[(1 + b\,\omega_b)^6 - 1\right] + p + \frac{p}{\omega_b}\left[(1 + b\,\omega_b\,\omega_{bp})^6 - 1\right]}. \tag{S4}$$

Following recent work (*Gregor et al., 2007*; *Park et al., 2019*), we now assume that the Bicoid-Bicoid pairwise cooperativity is very strong ($\omega_b \gg 1$). We can then simplify *Equation S4* to obtain

$$P_{bound} = \frac{p + p\,b^6\,\omega_b^5\,\omega_{bp}^6}{1 + p + b^6\,\omega_b^5 + p\,b^6\,\omega_b^5\,\omega_{bp}^6}. \tag{S5}$$

If we now define a new binding constant for Bicoid, $K_b' = K_b * (\frac{1}{\omega_b})^{\frac{5}{6}}$, such that $b' = b\,\omega_b^{\frac{5}{6}}$, and a new cooperativity term between Bicoid and RNAP given by $\omega_{bp}' = \omega_{bp}^6$, we can then rewrite *Equation S5* as

$$P_{bound} = \frac{p + b'^6\,p\,\omega_{bp}'}{1 + p + b'^6 + b'^6\,p\,\omega_{bp}'}, \tag{S6}$$

which is the expression we use throughout the main text. Thus, strong pairwise cooperativity between Bicoid molecules leads to a functional form where only the state with all Bicoid molecules bound remain (six in this case). This strong cooperativity can explain the sharp step-like expression pattern along the embryo's anterior-posterior axis of the *hunchback* gene (*Figure 3J*; *Gregor et al., 2007*, *Park et al., 2019*, *Driever and Nüsslein-Volhard, 1988*; *Driever and Nüsslein-Volhard, 1989*).

## S2 Derivation of the general and simpler thermodynamic model for the *hunchback* P2 enhancer with one Runt binding site

Having derived the equation for the strong cooperative binding of Bicoid to the wild-type *hunchback* P2 enhancer, we will now extend that model to the case of *hunchback* P2 with one Runt binding site. The corresponding states and weights of our full model are shown in *Figure 2—figure supplement 2A*.

Using a similar logic for calculating the partition functions as described in the previous section, we can compute the probability of the promoter being bound by RNAP as

$$
\text{p}_{bound} = \frac{\overbrace{\left(p + p\sum_{i=1}^{6}\binom{6}{i}b^{i}\omega_b^{i-1}\omega_{bp}^{i}\right)}^{\text{Bicoid and RNAP}} + \overbrace{\left(r\,p\,\omega_{rp} + r\,p\,\omega_{rp}\sum_{i=1}^{6}\binom{6}{i}b^{i}\omega_b^{i-1}\omega_{bp}^{i}\right)}^{\text{Bicoid, Runt, and RNAP}}}{\underbrace{\left(1+\sum_{i=1}^{6}\binom{6}{i}b^{i}\omega_b^{i-1}\right)}_{\text{Bicoid only}} + \underbrace{\left(p + p\sum_{i=1}^{6}\binom{6}{i}b^{i}\omega_b^{i-1}\omega_{bp}^{i}\right)}_{\text{Bicoid and RNAP}} + \underbrace{\left(r + r\sum_{i=1}^{6}\binom{6}{i}b^{i}\omega_b^{i-1}\right)}_{\text{Bicoid and Runt}} + \underbrace{\left(rp\omega_{rp} + rp\omega_{rp}\sum_{i=1}^{6}\binom{6}{i}b^{i}\omega_b^{i-1}\omega_{bp}^{i}\right)}_{\text{Bicoid, Runt, and RNAP}}}, \tag{S7}
$$

where, in addition to the parameters defined in the above section for the wild-type *hunchback* P2 case in the absence of Runt, we have added two parameters: the dissociation constant for Runt given by $K_r$, and a Runt-RNAP interaction term (an anti-cooperativity), $\omega_{rp}$. Using the binomial theorem as in *Equation S2*, we can simplify *Equation S7* to obtain

$$
\text{p}_{bound} = \frac{p + \frac{p}{\omega_b}\left[\left(1+b\omega_b\omega{bp}\right)^6-1\right]+rp\omega_{rp}+\frac{rp\omega_{rp}}{\omega_b}\left[\left(1+b\omega_b\omega_{bp}\right)^6-1\right]}{1+\frac{1}{\omega_b}\left[\left(1+b\omega_b\right)^6-1\right]+p+\frac{p}{\omega_b}\left[\left(1+b\omega_b\omega{bp}\right)^6-1\right]+r+\frac{r}{\omega_b}\left[\left(1+b\omega_b\right)^6-1\right]+rp\omega_{rp}+\frac{rp\omega_{rp}}{\omega_b}\left[\left(1+b\omega_b\omega_{bp}\right)^6-1\right]}. \tag{S8}
$$

We now again assume that Bicoid-Bicoid cooperativity is very strong such that $\omega_b \gg 1$. Then, we can combine *Equation S8* with *Equation 1* to obtain

$$
Rate = R\,\text{p}_{bound} = R\,\frac{p + b'^{6}\,p\,\omega_{bp} + r\,p\,\omega_{rp} + b'^{6}\,r\,p\,\omega'_{bp}\,\omega_{rp}}{1+b'^{6}+r+b'^{6}\,r+p+b'^{6}\,p\,\omega'_{bp}+r\,p\,\omega_{rp}+b'^{6}\,r\,p\,\omega'_{bp}\,\omega_{rp}}, \tag{S9}
$$

where the new parameters, $b'$ and $\omega'_{bp}$ are defined in the same way as in *Equation S6*. The effective states and weights remaining after taking this limit are shown in *Figure 2—figure supplement 2B*. Similarly, we can derive expressions for $p_{bound}$ in the presence of two and three Runt binding sites, and in the strong Bicoid-Bicoid cooperativity limit in order to obtain the predictions used throughout this text. We show this expression for two Runt binding sites in *Equation 6*. Further, for the case of repression by three Runt binding sites, the rate of transcription is given by

$$
\begin{aligned}
Rate = \quad & R\left(p + b^{6}p\omega_{bp} + rp(\omega_{rp1}+\omega_{rp2}+\omega_{rp3}) + b^{6}rp\omega_{bp}(\omega_{rp1}+\omega_{rp2}+\omega_{rp3})+\right.\\
& r^{2}p(\omega_{rp1}\omega_{rp2}\omega_{rr1}\omega_{rrp1} + \omega_{rp2}\omega_{rp3}\omega_{rr2}\omega_{rrp2} + \omega_{rp3}\omega_{rp1}\omega_{rr3}\omega_{rrp3})+\\
& r^{3}p\omega_{rp1}\omega_{rp2}\omega_{rp3}\omega_{rr1}\omega_{rr2}\omega_{rr3}\omega_{rrp1}\omega_{rrp2}\omega_{rrp3}\omega_{rrr}\omega_{rrrp}+\\
& b^{6}r^{2}p\omega_{bp}(\omega_{rp1}\omega_{rp2}\omega_{rr1}\omega_{rrp1} + \omega_{rp2}\omega_{rp3}\omega_{rr2}\omega_{rrp2} + \omega_{rp3}\omega_{rp1}\omega_{rr3}\omega_{rrp3})+\\
& \left. b^{6}r^{3}p\omega_{bp}\omega_{rp1}\omega_{rp2}\omega_{rp3}\omega_{rr1}\omega_{rr2}\omega_{rr3}\omega_{rrp1}\omega_{rrp2}\omega_{rrp3}\omega_{rrr}\omega_{rrrp}\right)\\
& \left(1 + b^{6}(1+3r+p\omega_{bp}) + 3r + p + rp(\omega_{rp1}+\omega_{rp2}+\omega_{rp3})+\right.\\
& r^{2}p(\omega_{rp1}\omega_{rp2}\omega_{rr1}\omega_{rrp1} + \omega_{rp2}\omega_{rp3}\omega_{rr2}\omega_{rrp2}\omega_{rp3}\omega_{rp1}\omega_{rr3}\omega_{rrp3}) + b^{6}r^{2}(\omega_{rr1}+\omega_{rr2}+\omega_{rr3})+\\
& b^{6}r^{3}\omega_{rr1}\omega_{rr2}\omega_{rr3}\omega_{rrr} + r^{2}(\omega_{rr1}+\omega_{rr2}+\omega_{rr3}) + r^{3}\omega_{rr1}\omega_{rr2}\omega_{rr3}+\\
& r^{3}p\omega_{rp1}\omega_{rp2}\omega_{rp3}\omega_{rr1}\omega_{rr2}\omega_{rr3}\omega_{rrp1}\omega_{rrp2}\omega_{rrp3}\omega_{rrr}\omega_{rrrp}+\\
& b^{6}rp\omega_{bp}(\omega_{rp1}+\omega_{rp2}+\omega_{rp3}) + b^{6}r^{2}p\omega_{bp}\left(\omega_{rp1}\omega_{rp2}\omega_{rr1}\omega_{rrp1} + \omega_{rp2}\omega_{rp3}\omega_{rr2}\omega_{rrp2} + \omega_{rp3}\omega_{rp1}\omega_{rr3}\omega_{rrp3}\right)+\\
& \left. b^{6}r^{3}p\omega_{bp}\omega_{rp1}\omega_{rp2}\omega_{rp3}\omega_{rr1}\omega_{rr2}\omega_{rr3}\omega_{rrp1}\omega_{rrp2}\omega_{rrp3}\omega_{rrr}\omega_{rrrp}\right)^{-1},
\end{aligned} \tag{S10}
$$

where the parameters are defined as in *Figure 1* and Section 'Repression by three-Runt binding sites also requires higher-order cooperativity'.

## S3 Comparing using static versus dynamic transcription factor concentrations as model inputs

In this section, we tested whether using static, time-averaged transcription factor concentration profiles yielded comparable theoretical predictions than when instead acknowledging the fact that input transcription factor concentration changes over time. Briefly, we compared the predicted rate of transcription calculated in two ways: (1) time-averaging the instantaneous rate from the dynamic transcription factor concentration profiles over a specified time window (from 5 to 10 minutes from the 13th anaphase) and (2) using static input transcription factors already time-averaged over the same time window.

As a concrete example, we focused on the *hunchback* P2 enhancer with one Runt binding site. We calculated the predicted rate of transcription using the thermodynamic model given by *Equation 2*. First, we performed this calculation using the dynamic concentration profiles of Bicoid and Runt shown in *Figure 3B and D*, respectively. Briefly, the terns $b$ and $r$ in *Equation 2* now become functions of time such that

$$Rate(t) = R \; \frac{p + b^6(t) \, p \, \omega_{bp} + r(t) \, p \, \omega_{rp} + b^6(t) \, r(t) \, p \, \omega_{bp} \, \omega_{rp}}{1 + b^6(t) + r(t) + b^6(t) \, r(t) + p + b^6(t) \, p \, \omega_{bp} + r(t) \, p \, \omega_{rp} + b^6(t) \, r(t) \, p \, \omega_{bp} \, \omega_{rp}}, \tag{S11}$$

where $b(t) = [Bicoid](t)/K_b$ and $r(t) = [Runt](t)/K_r$. We choose a set of reasonable values for the model parameters to illustrate the calculation of $Rate(t)$ at 30% of the embryo length. The resulting dynamic rate of transcription profile is shown in *Figure 3—figure supplement 1A* (blue curve). We then use this profile to calculate the time-averaged rate of transcription over the time window of 5–10 minutes from the 13th anaphase, resulting in the green area shown in *Figure 3—figure supplement 1A*.

The predicted average rate of RNAP loading given dynamic input transcription factors can be compared to the predicted rate of RNAP loading given the average input concentrations that we used throughout the main text (*Figure 3E*). Specifically, we plug the static concentration profiles of Bicoid and Runt shown in *Figure 3E* into *Equation 2* to obtain the red area shown in *Figure 3—figure supplement 1A*. As shown in the figure, the predicted rate of transcription obtained by these two analysis methodologies are equivalent within error.

Finally, we performed this comparison between different approaches to calcualate the rate of transcription as a function of position along the embryo (from 20% to 70% of the embryo length). As shown in *Figure 3—figure supplement 1B*, the resulting spatial profiles are comparable within error. Thus, we have shown that our approach of using time-averaged, static transcription factor concentrations as inputs to our model yield quantitatively equivalent result as accounting for the dynamic concentration profiles of these transcription factors.

## S4 Quantitative interpretation of MS2 signals

The MS2 signal reports on three features of transcriptional dynamics: (1) the initial RNAP loading rate, (2) the duration of transcription, and (3) the fraction of loci that engage in transcription at any time point in the nuclear cycle. In this section, we will explain in further detail how we extract these features from the MS2 signal over nuclear cycle 14.

### S4.1 Extracting the initial RNAP loading rate

The initial rate of RNAP loading corresponds the average transcription rate observed after transcriptional onset and until the MS2 signal reaches its peak value during nuclear cycle 14. In order to measure this rate, we followed the protocol described in *Garcia et al., 2013*. Briefly, as shown in *Figure 3—figure supplement 2A*, we fitted a line to the MS2 time trace (averaged over nuclei within a spatial window of 2.5% of the embryo length) within the time window of 5–10 minutes after the 13th anaphase. The slope of this line reported on the initial rate of RNAP loading (*Figure 3G*). The spatial profiles of this initial rate of RNAP loading across all our synthetic enhancer constructs and genotypes are shown in *Figure 3—figure supplement 2B*.

### S4.2 Extracting the duration of transcription

In the main text, we focused on the theoretical prediction of the initial rate of transcription. However, the length of the time window over which transcription occurs (*Lammers et al., 2020*) is another regulatory knob that, in principle, Runt could modulate to dictate gene expression patterns. We sought to determine the duration of time over which transcription occurs to assess whether Runt affects not only the initial rate of transcription, but also the time window over which transcription

could initiate. To quantify the effective duration of transcription initiation, we resorted to the analysis methodology developed in *Garcia et al., 2013*. Briefly, we parametrized the MS2 signal decay regime—after transcription reaches its peak and becomes slower than the unloading rate (*Garcia et al., 2013*)—as an exponential decay (*Figure 3—figure supplement 3A*). Thus, we can describe the MS2 spot fluorescence trace in the decay regime as

$$Fluo(t) = Fluo_{max}e^{-(t-T_{peak})/\tau},$$ (S12)

where $T_{peak}$ represents the time point where the MS2 spot fluorescence reaches its peaks, $Fluo_{max}$ is the maximum level of fluorescence, and $\tau$ is the decay time.

Given the sometimes noisy MS2 traces (data not shown), we fitted an exponential curve to the more robust integral of the MS2 spot fluorescence over time from $T_{peak}$ to the end of nuclear cycle 14 as shown in *Figure 3—figure supplement 3B*. This quantity is proportional to the amount of mRNA produced between the integration bounds (*Garcia et al., 2013*). The resulting accumulated mRNA time trace is then fitted to the integrated form of *Equation S12*, which is given by

$$mRNA\ (t) = mRNA_{max}(1 - e^{(t-T_{peak})/\tau}),$$ (S13)

where $mRNA_{max}$ is the accumulated mRNA at the end of nuclear cycle 14.

The resulting profiles of the duration of transcription along the embryo for our all synthetic enhancer constructs are illustrated in *Figure 3—figure supplement 3C* in the presence and absence of Runt protein. As shown in the figure, this duration time is not significantly modulated by Runt repressor.

## S4.3 Calculation of the fraction of competent nuclei

Another quantity that could be modulated by Runt repressor is the fraction of loci that ever engage in transcription during a given nuclear cycle, which we termed as the "fraction of competent loci". As demonstrated by *Garcia et al., 2013*, *Dufourt et al., 2018*, *Lammers et al., 2020* and *Eck et al., 2020*, this fraction of transcriptionally competent loci is modulated along the anterior-posterior axis, presumably due to the action of transcription factor gradients.

To show a concrete example of how this quantity is calculated, we take data from one construct ([000]) showing the MS2 spot fluorescence time traces from individual loci of transcription as shown in *Figure 3—figure supplement 4A*. Here, columns represent time points during nuclear cycle 14, and rows represent individual transcriptional loci. As shown in the figure, roughly 80% of the loci, labeled as "competent loci", show active transcription during nuclear cycle 14. However, the remaining 20% of the loci never engage in transcription, which we termed as "incompetent loci". Because these two populations exhibit wildly different behaviors, we define the fraction of competent loci as

$$\text{fraction of competent loci} = \frac{\text{number of competent loci}}{\text{number of total loci}}.$$ (S14)

Thus, in this example in *Figure 3—figure supplement 4A*, the fraction of competent loci is approximately 0.8.

*Figure 3—figure supplement 4B* shows the measured fraction of active loci for all synthetic enhancer constructs in the presence and absence of Runt repressor. As seen in the figure, although this quantity can be modulated by the presence of Runt repressor, this is not always the case (e.g., [011] and [111]). Moreover, we could not find a trend for how the fraction of competent loci is modulated by different combinations of Runt binding sites. For example, the [100] construct alone did show a change in the fraction of active loci in the presence of Runt, whereas the [010] construct did not. When these two binding sites were combined as the [110], there was no significant modulation of the fraction of competent loci when adding Runt repressor. In another example, the [001] construct showed a mild modulation of the fraction of competent loci. However, when this Runt binding site was combined with the [010], which did not show any modulation, the [011] construct showed a much bigger modulation of the fraction of competent loci than the [001]. Thus, the [010] Runt binding site could drive more or less modulation of the fraction of competent loci when combined with different Runt binding sites in a context-dependent manner. As a result of our failure to uncover an apparent trend in terms of which regulatory architectures lead to a stronger

modulation of the fraction of active loci, we did not attempt to theoretically explain the regulation of this fraction of active loci in this study.

## S5 Design of synthetic enhancer constructs based on the *hunchback* P2 enhancer

The Runt binding sites were introduced into the *hunchback* P2 minimal enhancers at the positions determined by *Chen et al., 2012*. To make this possible, the authors chose positions containing presumed neutral DNA sequences, meaning that these DNA locations did not contain obvious motifs for Bicoid or Zelda, the major input transcription factors that regulate this enhancer. Then, these DNA sequences were mutated to turn them into Runt binding sites.

To ensure that this process did not perturb the binding sites for Bicoid and Zelda we resorted to the Advanced PATSER entry form (*Hertz et al., 1990*; *Hertz and Stormo, 1999*) which identifies the location of transcription factor binding sites from a sequence of DNA based on position weight matrices. We used position weight matrices for Bicoid and Zelda from *Park et al., 2019*. PATSER was run with the settings described in *Eck et al., 2020* for both the *hunchback* P2 enhancer and the *hunchback* P2 enchancer with three Runt binding sites (from *Chen et al., 2012*) for Bicoid and Zelda, respectively. The result of this analysis for these two constructs is shown for each transcription factor in *Figure 4—figure supplement 1A*. Here, we took a PATSER score cutoff—for considering a given sequence to be a binding site—of 3 as in *Eck et al., 2020*. We observed that the recognized binding motifs for both Bicoid and Zelda were identical between the two constructs, meaning that we did not add additional Bicoid or Zelda binding sites by introducing the Runt motifs. The resulting synthetic enhancer with three Runt binding sites with mapped binding sites for Bicoid, Zelda (*Figure 4—figure supplement 1A*), and Runt (*Chen et al., 2012*) is shown in *Figure 4—figure supplement 1B* as a reference. The position of the Runt binding sites are noted from their distance from the promoter (which is marked as 0).

## S6 Markov Chain Monte Carlo inference protocol

Markov Chain Monte Carlo (MCMC) sampling is a widely used technique for robust parameter estimation using Bayesian statistics (*Geyer and Thompson, 1992*; *Sivia and Skilling, 2006*). We used the MATLAB package *MCMCstat*, an adaptive MCMC technique, which we could directly implement downstream of our data analysis pipeline (*Haario et al., 2006*; *Haario et al., 2001*). Detailed instructions on how to implement the *MCMCstat* package can be found in https://mjlaine.github.io/mcmcstat/, (copy archived at swh:1:rev:ca0cbd288f03c7a29050b3d6698a96b45ccfa4b2; *Laine, 2021*).

MCMC allows for an estimation of the set of parameter values of a model that best explain the experimental data along with their associated errors. In this work, we used MCMC to infer the set of best fit values of the parameters in our thermodynamic models given the observed profile of the rate of transcription initiation along the anterior-posterior axis of the embryo.

MCMC calculates a Bayesian posterior probability distribution of each free parameter given the data by stochastically sampling different parameter values. For a given set of observations $D$ and a model with parameters $\theta$, the posterior probability distribution of a particular set of values is given by Bayes' theorem

$$\underbrace{p(\theta|D)}_{\text{posterior}} \propto \underbrace{p(D|\theta)}_{\text{likelihood}} \underbrace{p(\theta)}_{\text{prior}}. \tag{S15}$$

The prior function represents the a priori assumption about the probability distribution of parameter values $\theta$. Here, we assumed a uniform prior distribution for all parameters to reflect our ignorance about the model parameters within the following intervals:

- $K_b$: [0, 100] AU
- $\omega_{bp}$: [0, 200]
- $p$: [0, 1]
- $R$: [0, 400] AU/min
- $K_r$: [0, 100] AU
- $\omega_{rp}$: [0, 1.2]
- $\omega_{rr}$: [0, 100]
- $\omega_{rrp}$: [0, 100]

These intervals were justified using the following arguments.

First, because we observed a gradual modulation of the rate of transcription by both Bicoid and Runt in the middle region of the embryo we reasoned that the binding sites for these transcription factors were not saturated. As a result, we posited that the real dissociation constant should be between the minimum and maximum measured values of Bicoid and Runt (*Figure 3—figure supplement 2*). Our measurements of Bicoid and Runt concentration yield fluorescence values over the 0–100 AU range for the embryo region that we used for contrasting our model and experimental data (20–50% of the embryo length), such that the dissociation constants ($K_b$ and $K_r$) should not exceed the maximum value of the Bicoid or Runt concentration.

Second, $\omega_{bp}$ represents the cooperativity between Bicoid complex and RNAP. In the statistical mechanics framework, this cooperativity can be expressed using the interaction energy between Bicoid and RNAP, $\Delta\epsilon_{bp}$, such that $\omega_{bp} = exp(-\beta\Delta\epsilon_{bp})$, where $\beta = \frac{1}{k_B T}$, $k_B$ is the Boltzmann constant and $T$ is the temperature. There is not much known about in vivo interaction energies between Bicoid and RNAP complex, thus we tried several different bounds until we found a narrow enough parameter bound with unimodal distribution of the posterior chain. As we could see from the corner plots in *Figure 4C*, there is a positive correlation between $K_b$ and $\omega_{bp}$. Thus, we constrained the $\omega_{bp}$ intervals by finding an interval that gives both well-constrained $K_b$ and $\omega_{bp}$ (*Figure 4C*).

Third, $R$ represents the rate of RNAP loading when the promoter is occupied, thus it is constrained by the maximum observed rate of RNAP loading (*Figure 3—figure supplement 2*).

Fourth, $p = ([RNAP]/K_p)$ represents the concentration of RNAP divided by its dissociation constant. Recall that the predicted rate of transcription from *hunchback* P2 in the limit where the Bicoid concentration reaches zero is given by

$$Rate([Bicoid] \to 0) = R\,\frac{p}{1+p}. \tag{S16}$$

This rate of transcription at the posterior region, where Bicoid reaches zero, is much lower than that at the anterior region where Bicoid saturates given by $R$ (*Figure 3—figure supplement 2*). As a result, we can write the inequality

$$R\,\frac{p}{1+p} \ll R. \tag{S17}$$

such that

$$\frac{p}{1+p} \ll 1, \tag{S18}$$

which holds if $p \ll 1$.

Finally, we did not have good estimates for the intervals of either Runt-Runt cooperativity, $\omega_{rr}$, or higher-order cooperativity, $\omega_{rrp}$. Thus, we initially started with an interval of $[0, 100]$, of the same order as the interval we used $\omega_{bp}$. We then explored whether this parameter bound was sufficient to give us constrained values of $\omega_{rr}$ and $\omega_{rrp}$. As we showed in *Figure 6—figure supplement 6D*, this interval gives reasonably constrained values of $\omega_{rr}$ and $\omega_{rrp}$. As shown in *Figure 6* and *Figure 6—figure supplement 6*, we posit that the $\omega_{rr}$ parameter is not well-constrained not because of its width of the interval, but because it is not as essential for the model fit to the data as it is to include $\omega_{rrp}$ into the model. Overall, our MCMC inference results as well as the corner plots shown demonstrate that our parameter intervals chosen were reasonable.

## S6.1 Calculation of the Akaike Information Criterion

The Akaike Information Criterion (AIC) is a quantitative metric to assess how well a model performs compared to other models (*Akaike, 1974*). AIC is defined by two terms that account for: (1) how well the model explains the data (log-likelihood), and (2) how many parameters are used (to penalize the model). The AIC is expressed by

$$AIC = -2\log(\text{likelihood}) + 2p, \tag{S19}$$

where $p$ is the number of free parameters in the model. As a result, given a set of models, the model with minimal AIC value is preferred (*Akaike, 1974*).

First, we assume that our measurement error follows a Gaussian probability distribution. As a result, we can write down the probability of measuring a k-th datum, $x_k$ from a Gaussian distribution whose mean is $\mu_k$ and standard deviation is $\sigma_k$, termed $prob(x_k|\mu_k, \sigma_k)$(*Sivia and Skilling, 2006*)

$$prob(x_k|\mu_k, \sigma_k) = \frac{1}{\sigma_k\sqrt{2\pi}}exp(-\frac{(x_k-\mu_k)^2}{2\sigma_k^2}).$$ (S20)

We can write the likelihood $L$ of a given set of $N$ measurements by multiplying the probabilities of each independent measurement such that

$$L = prob(\{x_1, x_2, ..., x_N\}|\{\mu_1, \mu_2, ..., \mu_N\}, \{\sigma_1, \sigma_2, ..., \sigma_N\}) = \Pi_{k=1}^{N}\frac{1}{\sigma_k\sqrt{2\pi}}exp(-\frac{(x_k-\mu_k)^2}{2\sigma_k^2}),$$ (S21)

where $x_k$ is the k-th datum with standard deviation of $\sigma_k$ from an expected value of $\mu_k$ from a model (*Sivia and Skilling, 2006*).

By taking the logarithm of the likelihood (L) we get

$$\log(L) = -\sum_{k=1}^{N}\log(\sigma_k\sqrt{2\pi}) - \sum_{k=1}^{N}\frac{(x_k-\mu_k)^2}{2\sigma_k^2}.$$ (S22)

We can now plug in this expression for the log-likelihood into the definition of the AIC from *Equation S19*, resulting in

$$AIC = 2k + 2\sum_{k=1}^{N}(\sigma_k\sqrt{2\pi}) + 2\sum_{k=1}^{N}\frac{(x_k-\mu_k)^2}{2\sigma_k^2}.$$ (S23)

All of the Akaike Information Criteria values shown in *Figure 6—figure supplement 1* and *Figure 5—figure supplement 3* are calculated using *Equation S23*.

## S7 Comparison of different modes of repression

Transcriptional repressors have been classified into two broad categories: short-range and long-range, depending on the genomic length scale that they act on *Courey and Jia, 2001*; *Li and Gilmour, 2011*. Long-range repression is realized by the recruitment of chromatin modifiers. In contrast, short-range repressors act within 100–150 bp by interacting with nearby transcription factors or with the promoter (*Li and Gilmour, 2011*). Traditionally, the molecular mechanism of short-range repressors, such as Runt, have been further classified into three categories: "direct repression", "competition", and "quenching" (*Gray et al., 1994*; *Jaynes and O'Farrell, 1991*; *Arnosti et al., 1996*; *Kulkarni and Arnosti, 2005*). In "direct repression", the repressor inhibits the binding of RNAP to the promoter (*Figure 5—figure supplement 1A*). "Competition" denotes a repressor that competes with an activator for the same DNA binding location (*Figure 5—figure supplement 1B*). This molecular mechanisms has been proposed for the action of Giant and Krüppel repressors on the *even-skipped* stripe 2 enhancer, where some activator and repressor binding sites partially overlap (*Small et al., 1992*). Lastly, "quenching" corresponds to the case where the repressor and activator do not interact with each other directly. Instead, the repressor inhibits the activators' action of recruiting the RNAP (*Figure 5—figure supplement 1C*).

Despite several classic studies of the molecular mechanism of repressors in the early fly embryo (*Gray et al., 1994*; *Ip et al., 1992*; *Bothma et al., 2011*; *Jaynes and O'Farrell, 1991*), the mechanisms of many repressors remain unknown. Note that, even for the same repressor, the mode of repression might not be the same depending on, for example, its sequence context (*Koromila and Stathopoulos, 2019*; *Hang and Gergen, 2017*). For example, it has been proposed that Runt repressor acts with different mechanisms in different regulatory elements of the *sloppy-paired* gene (*Hang and Gergen, 2017*). In this section, we derive a thermodynamic model from each mode of repression and compare their explanatory power in the context of our data stemming from the *hunchback* P2 enhancer containing one Runt binding site. Note that, in the main text, we already developed a thermodynamic model for the "direct repression" scenario (Section 'Derivation of the general and simpler thermodynamic model for the hunchback P2 enhancer with one Runt binding site'). As a result, in this section, we focus on deriving the thermodynamic models for the "competition" and "quenching" scenarios, but repeat the result of the derivation for the "direct repression" here for ease comparison between different models.

## S7.1 Derivation of models for each scenario of repression for *hunchback* P2 with one Runt binding site

### S7.1.1 Modeling repression for *hunchback* P2 with one Runt binding site: direct repression

For completeness, we repeat the expression for the direct repression scenario as shown in Section 'Derivation of the general and simpler thermodynamic model for the *hunchback* P2 enhancer with one Runt binding site' and *Figure 5—figure supplement 1A*. The probability of finding RNAP bound to the promoter, $p_{bound}$, is calculated by dividing the sum of all statistical weights featuring RNAP by the sum of the weights of all possible microstates. The calculation of $p_{bound}$, combined with *Equation 1*, leads to the expression

$$Rate = R\, p_{bound} = R\, \frac{p+b^6\, p\, \omega_{bp}+r\, p\, \omega_{rp}+b^6\, r\, p\, \omega_{bp}\, \omega_{rp}}{1+b^6+r+b^6\, r+p+b^6\, p\, \omega_{bp}+r\, p\, \omega_{rp}+b^6\, r\, p\, \omega_{bp}\, \omega_{rp}}, \tag{S24}$$

where the parameters are as defined in *Figure 2*.

### S7.1.2 Modeling repression for *hunchback* P2 with one Runt binding site: competition

In the competition scenario, Runt binding makes Bicoid binding less likely. This mechanism can be captured by an interaction term between Bicoid and Runt given by $\omega_{br}$. Building on our assumption of strong Bicoid-Bicoid cooperativity, we posit that Runt disfavors the state with six bound Bicoid molecules. We can enumerate the states and weights from *Figure 5—figure supplement 1B* to calculate the *Rate* ($\propto p_{bound}$), which leads to

$$Rate = R\, \frac{p+b^6\, p\, \omega_{bp}+r\, p+b^6\, r\, p\, \omega_{bp}\, \omega_{br}}{1+p+b^6+b^6\, r\, \omega_{br}+b^6\, p\, \omega_{bp}+r\, p+b^6\, r\, p\, \omega_{bp}\, \omega_{br}}. \tag{S25}$$

### S7.1.3 Modeling repression for *hunchback* P2 with one Runt binding site: quenching

In the quenching scenario, Runt reduces the magnitude of the cooperativity between the Bicoid complex and RNAP by a factor $\omega_{brp}$. We can enumerate the states and weights from *Figure 5—figure supplement 1C*, leading to a rate of transcription given by

$$Rate = R\, \frac{p+b^6\, p\, \omega_{bp}+r\, p+b^6\, r\, p\, \omega_{bp}\, \omega_{brp}}{1+p+b^6+r+b^6\, r+b^6\, p\, \omega_{bp}+r\, p+b^6\, r\, p\, \omega_{bp}\, \omega_{brp}}. \tag{S26}$$

With these expressions for each repression mechanism in hand, we can now compare how each model fares against our experimental data.

## S7.2 Comparing the three models of repression with the one-Runt binding site data

We used the MCMC sampling to fit each model to our experimentally measured initial rate of transcription over the anterior-posterior axis of the embryo. As shown in *Figure 5—figure supplement 2A, B and C*, we see that all three models can explain the [100] and [010] construct data relatively well. However, the competition model resulted in a qualitatively poor fit to the [001] construct as shown by the lack of saturation in the most anterior region of the embryo (*Figure 5—figure supplement 2C*), (ii). The direct repression and quenching models showed equally good fits to the data stemming from this construct.

## S7.3 Predicting two-Runt binding sites data for each mode of repression

We further tested these different models of repression by using the parameters inferred from the one-Runt binding site constructs to predict the rate of initiation for the two-Runt binding sites constructs. As reasoned in the main text, we began by assuming that the two Runt molecules act independently of each other such that there are no interactions between Runt molecules. *Figure 6—figure supplement 1* shows this parameter-free prediction for our two-Runt binding sites constructs for all three modes of repression. As shown in the figure, none of the models can explain the data, suggesting the need to invoke additional interactions between the molecular players of our model.

Next, we considered whether Runt-Runt pairwise or higher-order cooperativities had to be invoked in order to explain the two-Runt binding sites data for both the competition and quenching mechanisms. For the competition model, we considered Runt-Runt cooperativity, $\omega_{rr}$, and Runt-Runt-Bicoid higher-order cooperativity, $\omega_{brr}$ in addition to the Runt-Bicoid interaction term $\omega_{br}$. In the quenching scenario, we accounted for Runt-Runt cooperativity, $\omega_{rr}$, and Runt-Runt-Bicoid-RNAP higher-order cooperativity, $\omega_{brrp}$. For both the competition (*Figure 6—figure supplement 2*) and quenching (*Figure 6—figure supplement 3*) mechanisms, we observed a qualitatively similar trend to that observed for direct repression (*Figure 6*). Specifically, as shown in *Figure 6—figure supplements 2C and 3C*, considering pairwise cooperativity did not significantly improve the MCMC fits to the data for either model considered. Further, considering only the higher-order cooperativity also did not improve the fits for both competition and quenching mechanisms as shown in *Figure 6—figure supplement 2D* and *Figure 6—figure supplement 3D*. Invoking both Runt-Runt cooperativity and higher-order cooperativity improved the fits qualitatively for both competition and quenching mechanisms as shown in *Figure 6—figure supplement 2E* and *Figure 6—figure supplement 3E*.

While the quenching model showed almost equally good MCMC fits to the data as the direct repression model, the competition model showed qualitatively poor fits in any combination of cooperativities. In particular, there was a significant mismatch in the most anterior region of the embryo, where Bicoid is thought to saturate *hunchback* expression. While we do not view these fits as conclusive evidence to support one mechanism over the other, an exercise that would require a new round of experimentation, we conclude that higher-order cooperativity is required to explain the data from the two-Runt binding sites constructs regardless of the choice of mechanism of Runt.

## S8 Quantifying the nuclear concentration of LlamaTag-Runt

The major caveat in the eGFP:LlamaTag-Runt fluorescence measurements is that the raw nuclear fluorescence that we measured consists of two populations: eGFP *bound* to the LlamaTag-Runt, and *free*, *unbound* eGFP. Thus, in order to measure nuclear Runt concentration, we need to factor out the contribution from free eGFP to the overall fluorescence.

We followed the procedure described in *Bothma et al., 2018* which consists of using cytoplasmic fluorescence to calculate the free nuclear eGFP under two assumptions. First, we posit that most of the transcription factors reside in the nucleus such that the cytoplasmic fluorescence mostly reports on free cytoplasmic eGFP. Second, we assume that the nucleus-to-cytoplasm ratio of free eGFP is kept constant at a measured chemical equilibrium of $K_G = GFP_C/GFP_N = 0.8$, where $GFP_C$ and $GFP_N$ are the eGFP fluorescence in nuclei and cytoplasm in the absence of LlamaTag (*Bothma et al., 2018*).

As shown in *Bothma et al., 2018*, the nuclear concentration of the GFP-tagged transcription factor, $GFP - TF_N$, is given by

$$GFP - TF_N = Fluo_N - \frac{Fluo_C}{K_G}, \tag{S27}$$

where $Fluo_N$ and $Fluo_C$ are the eGFP fluorescence in nuclei and cytoplasm, respectively, that we measured in the embryos with both eGFP and LlamaTagged Runt. The resulting nuclear concentration of LlamaTag-Runt is shown in *Figure 3B*.

