## [Editor Report]

The work by Kim et al., used synthetic constructs in *Drosophila* to examine the relationship between regulators and transcription initiation. By measuring regulator concentrations and the corresponding RNA polymerase initiation rates in different synthetic constructs and using a thermodynamic model, the authors concluded that higher-order cooperativities between the repressor on adjacent binding sites, and that between the repressor and RNA polymerase are needed to explain the observed response curves in RNA polymerase loading rate. This work targets a challenging question in eukaryotic transcription regulation, where higher-order cooperativity between different molecular components, in addition to simple transcription factor binding and unbinding, is often necessary to account for observed promoter behaviors when multiple elements (repressors, mediators, activators) exist.

---

## [Decision Letter]

**Decision letter after peer review:**

Thank you for submitting your article "Predictive modeling reveals that higher-order cooperativity drives transcriptional repression in a synthetic developmental enhancer" for consideration by *eLife*. Your article has been reviewed by 3 peer reviewers, and the evaluation has been overseen by a Reviewing Editor and Naama Barkai as the Senior Editor. The reviewers have opted to remain anonymous.

The reviewers have discussed their reviews with one another, and the Reviewing Editor has drafted this to help you prepare a revised submission. While the work is potentially important, the uniqueness of the model and experimental verification of model predictions need to be further justified.

The essential revisions are:

1. Modeling: Please provide a better articulation/construction of the model regarding the fitting parameters (which to keep constant and which to vary, the number of parameters to use, error bars in the experimental data), whether an alternative model should be considered, whether the provided model can indeed account quantitatively observed data (not just to demonstrate that previous models do not work), and whether statistically the model fits experimental data adequately (not by overfitting, examine using Akaike Information Criterion AIC or Bayesian Information Criterion BIC). See reviewers #1 pt 1, #2 pt 1-5, and #3's overall review, and pt1.

2. Experiments: please provide experimental measurement for a few conditions where the outcome of transcription regulation by Runt is measured by smFISH or protein concentrations to demonstrate that the measured promoter activity using RNAP loading rate is equivalent or better (without complications in mRNA/protein degradation).

*Reviewer #1 (Recommendations for the authors):*

1. Modeling: I would like to see a better articulation of what the alternative is to a 'thermodynamic model.' I understand the authors meaning to be equilibrium models arrived at through partition functions. But kinetic models based on detailed balance are also thermodynamic. I don't want to get twisted around in the language, but the over-emphasis on the ideological purity of thermodynamics was distracting and seems to refer to an ongoing dialog / conversation which is likely lost on the general reader.

a. One example is when they assume that the Runt dissociation constant remains unchanged with different positions (K_r_) because the sequence is the same. The problem stems from their previous section where they found that there was an unsuspected relationship between the position of the Runt Binding sites and the parameter "p=[RNAP]/K_p_" --- the point being that the sequence is the same in each construct, and therefore, based on their model K_p_ is changing. So why is it preferrable to assume that K_r_ is constant with position, but K_p_ changes?

b. Expanding on the previous point, they then proceed to state that an unchanging K_r_ is supported by the fact that by only varying the w_rp_ they were able to fit their data. Considering the similar role of K_r_ with w_rp_ within the equation, I wonder if one could reach the same result varying the K_r_ and holding w_rp_ constant? Furthermore, who's to say that both don't vary between the constructs? Could the authors address this concern by varying K_r_ and further discussing this possibility and what it means for the interpretations of the model? In summary, the authors have a model which explains their data. Great. I'm not sure this agreement can be used as an argument for the propriety of the 'thermodynamic approach.'

2. A simple plot showing mRNA/cell as a function of repressor concentration for the different configurations in Figure 6 would be much appreciated. Perhaps I missed the inference, but the ultimate goal of an input/output understanding is to map concentration of a regulator to expression level of the target gene.

*Reviewer #2 (Recommendations for the authors):*

The manuscript has several issues as noted below.

1) For MCMC fitting of experimental data, the absolute error or relative error of all the results should be given. One would like to know the confidence degree of the fitting results.

2) Formula (5) in the main text is used to fit the data, and the results are shown in figure6. However, the number of the parameters in the fitting formula is almost the same as the amount of the experimental data, and the formula is too complicated. The corresponding parameter space is quite large. Therefore, it is necessary to further analyze the accuracy of the fitting and the necessity of adding the cooperativity.

3) What is source of error bar of the initial RNAP loading rate? Need to give detailed explanation. If the source of error bar is from the data of multiple traces, why not fit multiple trajectories separately?

4) The fitting equations (1), (2) and (5), are all the fitting function for the experimental data. Are the parameter b, p and r the fitting parameters? If they are the fitting parameters, the fitting result needs to be given. These parameters are related to the weights of different binding states in your model, which need to be listed in detail, and compared the differences of fitting results under variant conditions.

5) In this paper, a variety of data is shown, which are Runt binding to different sites such as [001][010],[011] and so on. Please elaborate on how Runt is experimentally controlled or determined to bind to the specified sites.

6) The fluorescence of nucleus and cytoplasm is quantitatively analyzed, how to distinguish nucleus and cytoplasm? Please give the process and corresponding results.

7) It is shown in Fiugre6b that if the cooperativity is not included, a good fitting result cannot be obtained. Has the fitting covered a large enough parameter space search? The fitting results in figure 6 are quite different from the experiment results regardless of the cooperativity. If so, does it mean that the cooperativity plays a major role in the fitting, or even that the individual interaction between Runt and binding sites can be ignored? Could you separate out the part of cooperativity in the fitting results so as to confirm that the cooperativity plays a key role in the experiment?

*Reviewer #3 (Recommendations for the authors):*

1) Results of Figure 5 should be accompanied by reports of fitting alternative models and suitable model comparisons performed. Also, a baseline model with constant K_R_ and \omega_RP_ (same for all three constructs) should be evaluated and compared to. Such comparison should ideally involve model comparison techniques such as likelihood ratio and AIC/BIC.

2) Results of Figure 6 should be accompanied by statistical assessment of improvements due to higher-order cooperativity parameters.

3) The fact that fitting of more complex models typically relied on setting different values for the additional parameters for each construct (rather than the same values for all tested constructs of that category) should be examined more closely. How well can the data be modeled without this flexibility?

4) The authors should address the concern that alternative model structures different from that used in the thermodynamics-based models here might have explained the data without requiring higher-order cooperativity.

Other suggestions:

One of the first experimental tests reported is that of predicted expression profile in the hunchback promoter with one Runt binding site. Figure 3J is supposed to be a qualitative match to the predictions in Figure 3F. Could the authors elaborate on what reasonable regimes of the model parameters would provide predictions that do not qualitatively match the observation? It seems like with an activator (Bicoid) progressively decreasing and a repressor (Runt) progressively increasing from anterior to posterior, the enhancer expression will obviously decrease in some form. Perhaps the authors might wish to make it clear that their goal here was to establish some basic parameter estimates to be used in the remaining analyses, and perhaps also to assess the uncertainties associated with those model parameters.

The results presented in Figure 4 are puzzling. Figure 4D suggests that the differences among the constructs arises purely from differences in the p and R parameters, and not from the bicoid or bicoid-RNAP related parameters. Could the authors add a possible mechanistic speculation here regarding how making changes solely in the enhancer sequence might result in biochemical changes solely in the behavior of RNAP at the promoter (without any change in the protein-DNA interactions at the enhancer itself)?

[Editors' note: further revisions were suggested prior to acceptance, as described below.]

Thank you for resubmitting your work entitled "Predictive modeling reveals that higher-order cooperativity drives transcriptional repression in a synthetic developmental enhancer" for further consideration by *eLife*. Your revised article has been evaluated by Naama Barkai (Senior Editor) and a Reviewing Editor.

The manuscript has been significantly improved but please further address one issue on the validation of transcription rate measurement, the reviewers asked to use an orthogonal method such as smFISH to validate the measurement using MS2, which was not done in the revision. Please provide at least a thorough comparison and contrast of the current method with other orthogonal methods in the Discussion to equip readers with a full context of the quantifications that are central to the work.

---

## [Author Response]

The essential revisions are:1. Modeling: Please provide a better articulation/construction of the model regarding the fitting parameters (which to keep constant and which to vary, the number of parameters to use, error bars in the experimental data), whether an alternative model should be considered, whether the provided model can indeed account quantitatively observed data (not just to demonstrate that previous models do not work), and whether statistically the model fits experimental data adequately (not by overfitting, examine using Akaike Information Criterion AIC or Bayesian Information Criterion BIC). See reviewers #1 pt 1, #2 pt 1-5, and #3's overall review, and pt1.

We apologize for the lack of clarity in our original manuscript regarding the model construction as well as the statistical interpretation of the MCMC inference. In our revised manuscript, we colored the free parameters in red to emphasize which parameters are being inferred from a given model equation. We also included error bars representing 95% of confidence intervals for all of our MCMC inference results in the main Figures (Figure 4, 5, 6, and 7).

For the question of whether an alternative model should be considered, specifically for one-Runt binding site constructs, we tested alternative models under four different scenarios: (1) K_r_ and w_rp_ being constant across constructs, (2) K_r_ remains constant and w_rp_ varies, (3) K_r_ varies and w_rp_ remains constant, and (4) both K_r_ and w_rp_ vary across constructs. The results are given in Figure S16, where we can see that our original assumption of keeping K_r_ constant and varying w_rp_ indeed explains the data better than the other models. This result is further supported by the quantification of model performance using the Akaike Information Criterion (as suggested by the reviewers) shown in Figure S16F. Finally, the inferred parameters from the case where we let both K_r_ and w_rp_ vary across constructs are shown in Figure S16G. These results suggest that the K_r_ values do not vary significantly across the constructs whereas w_rp_ values vary significantly between the constructs. From the aforementioned evidence, we argue that our assumption of constant K_r_ across constructs is well supported.

To assuage the reviewers’ concern on whether the improved model with higher-order cooperativities considered in the main text indeed explains the data better than the others, we computed Akaike Information Criterion for all models presented in Figure 6, resulting in a new panel, Figure 6G. The result indeed supports that both Runt-Runt cooperativity and higher-order cooperativity (emphasis on the higher-order cooperativity) are required to explain the experimental data without overfitting.

Finally, in addition to the “direct repression” that was assumed throughout the main text, we tested alternative models based on different modes of repression that have been proposed in the literature (Gray et al., Genes and Development 8:1829, 1994,). Specifically, we tested two additional modes of repression, “competition” and “quenching” in Figure S7 and S8, respectively, and summarized the results in Section S5. In short, our goal was to see if the pairwise cooperativities are sufficient to explain the two-Runt binding sites cases in these alternate models of repression, or whether we had to invoke higher-order cooperativities as in the “direct repression” mechanism. We concluded that both modes of repression required higher-order cooperativities to explain the two-Runt binding sites cases (even though the competition mechanism generally gave worse fits than direct repression or quenching). Thus, we believe that our conclusion of the necessity of higher-order cooperativity to account for our experimental data holds regardless of the specific model of repression that is assumed.

2. Experiments: please provide experimental measurement for a few conditions where the outcome of transcription regulation by Runt is measured by smFISH or protein concentrations to demonstrate that the measured promoter activity using RNAP loading rate is equivalent or better (without complications in mRNA/protein degradation).

We have revised our manuscript to include the results for accumulated mRNA patterns. Here is the summary of our major revisions.

First, we quantified the accumulated mRNA level (pattern) by integrating the MS2 time traces during nuclear cycle 14. This method was previously validated to give quantitatively comparable results to single-molecule fluorescence in situ hybridization (smFISH) (Garcia et al., Current Biology, 23:2140, 2013). The patterns of accumulated mRNA corresponding to each construct with and without the Runt protein are given in Figure S17. The figure clearly shows that the effect of repression is present at the level of accumulated mRNA.

Second, to demonstrate that the measured promoter activity using RNAP loading rate is equivalent to measures of accumulated mRNA, we compared the initial rate of RNAP loading to the accumulated mRNA at each position along the anterior-posterior axis for all of our constructs. The result is now included in a new figure, Figure S19. Interestingly, there is substantial correlation between these two metrics, with an average Pearson correlation coefficient of 0.9. Thus, we argue that the initial rate of RNAP loading, the quantity that we focused on in this paper, is comparable to the accumulated mRNA, suggesting that higher-order cooperativity is necessary to predict patterns of accumulated mRNA.

Regardless, we would like to emphasize that, even though there was a good correlation between the rate of transcription and the accumulated mRNA as shown in Figure S19, the objective of our work was not to predict accumulated mRNA patterns. Instead, our goal was to uncover the molecular mechanisms by which multiple transcription factors work together and to quantitatively predict the consequences of these mechanisms. We argue that the action of transcription factors is more accurately captured by monitoring the rate of transcription, rather than the accumulated mRNA, which could be potentially confounded by the regulation of different aspects of the transcription cycle (initiation, elongation and termination), as well as degradation and diffusion. We now emphasized this point in the Discussion (Lines 417-422).

Note that the reviewer requested to make this comparison using orthogonal techniques such as smFISH. However, we hope that the reviewers will agree that the already established relationship between the integral of the MS2 signal and the accumulated mRNA makes invoking such orthogonal techniques (which lack the temporal resolution of the MS2 system) unnecessary.

Reviewer #1 (Recommendations for the authors):1. Modeling: I would like to see a better articulation of what the alternative is to a 'thermodynamic model.' I understand the authors meaning to be equilibrium models arrived at through partition functions. But kinetic models based on detailed balance are also thermodynamic. I don't want to get twisted around in the language, but the over-emphasis on the ideological purity of thermodynamics was distracting and seems to refer to an ongoing dialog / conversation which is likely lost on the general reader.

We agree with the reviewer’s comment. Kinetic models that account for the transition rates between the different binding states the enhancer become thermodynamic models in the extreme case where the rates that characterize these transitions are much faster than the rate of mRNA production. In this limit, we can invoke quasi-equilibrium, making it possible to forego rates altogether and focus on the probabilities of each state. As the reviewer suggests, we could have entertained such a more general kinetic model. However, our goal was to ask whether the simple thermodynamic model, which has been very successful in bacteria, could also predict transcriptional regulation in the animal context. We have now included a discussion of alternative models in the main text (Lines 107-108) and Discussion (Lines 492-493) to clarify our points.

a. One example is when they assume that the Runt dissociation constant remains unchanged with different positions (K_r_) because the sequence is the same. The problem stems from their previous section where they found that there was an unsuspected relationship between the position of the Runt Binding sites and the parameter "p=[RNAP]/K_p_" --- the point being that the sequence is the same in each construct, and therefore, based on their model K_p_ is changing. So why is it preferrable to assume that K_r_ is constant with position, but K_p_ changes?

We agree that the justification for the assumption that the Runt dissociation constant K_r_ is equivalent throughout our constructs should be clarified further.

In Section 2.3 and Figure 4, we attempted to determine whether the simple thermodynamic model derived in Equation 3 could explain our data, and if so, which parameters were consistent across our synthetic enhancer constructs, and which parameters varied. As the reviewer pointed out, the parameter “*p=[RNAP]/K_p_*” changes between constructs, thus suggesting that K_p_ changes even though the sequence of the promoter is exactly the same across constructs. We do not know why this is the case, but we could speculate on a potential scenario.

We speculate that, since the enhancer is proximal to the promoter, the overall enhancer sequence might affect the binding of the transcriptional machinery itself. Therefore, the transcriptional machinery might see slightly different DNA sequences, a combination of the promoter and part of synthetic enhancer for each construct. Indeed, Louder et al. (Nature 531:604, 2018), showed using cryo-EM that promoter-bound general transcription factor IID (TFIID) occupies roughly 60 bp of DNA (from -20 to +40, where 0 represents the start of the transcription site in the promoter) when it forms pre-initiation complex. Thus, it is conceivable that the general transcriptional machinery would interact with the general DNA sequence of an enhancer located proximally upstream from the promoter. In contrast, the binding site sequence for Runt is small enough (8 bp) compared to the enhancer sequence (~300 bp) that Runt protein might see a similar local DNA sequence for each binding site. Thus, our results could be explained if the general transcriptional machinery interacts with much longer DNA sequences than Runt repressor. This discussion is now included in the Discussion (Lines 433-440)

Regardless of the reason behind the variation in K_p_ between reporter constructs, our assumption that the Runt dissociation constant remains equivalent across all binding sites is supported by relaxing this assumption and performing MCMC inference on an unconstrained model. In a new figure, Figure S16G, we show the inferred parameters corresponding to the case where we let both K_r_ and w_rp_ vary. In Author response image 1, it is clear that the inferred K_r_ values do overlap between constructs within error bars, whereas the p=[RNAP]/K_p_ varies across synthetic enhancer constructs. Thus, we believe that our assumption of K_r_ remaining constant across all of our constructs is valid, and chose to continue making this assumption throughout the main text. We have incorporated this evidence that the inferred K_r_ values do overlap between constructs within error bars in Figure S16.

**Author response image 1. sa2fig1:** Inferred values of K_r_ and [RNAP]/K_p_ across all synthetic enhancer constructs. K_r_ is inferred from one-Runt binding site constructs, and [RNAP]/K_p_ is inferred from all constructs in the absence of Runt protein.

b. Expanding on the previous point, they then proceed to state that an unchanging K_r_ is supported by the fact that by only varying the w_rp_ they were able to fit their data. Considering the similar role of K_r_ with w_rp_ within the equation, I wonder if one could reach the same result varying the K_r_ and holding w_rp_ constant? Furthermore, who's to say that both don't vary between the constructs? Could the authors address this concern by varying K_r_ and further discussing this possibility and what it means for the interpretations of the model? In summary, the authors have a model which explains their data. Great. I'm not sure this agreement can be used as an argument for the propriety of the 'thermodynamic approach.'

This is a great suggestion. We tested alternative models under four different scenarios: (1) K_r_ and w_rp_ being constant across constructs, (2) K_r_ remains constant and w_rp_ varies, (3) K_r_ varies and w_rp_ remains constant, and (4) both K_r_ and w_rp_ vary across constructs. The results are given in Figure S16, where we can see that our original assumption of keeping K_r_ constant and varying w_rp_ indeed explains the data better than the other models. Briefly, the best MCMC fits to our data look qualitatively similar for the [001] and [010] constructs regardless of our assumptions, but differ for the [100] construct. For the [100] construct, the best MCMC fits predict that there should be a repression in the presence of Runt protein for the (1) and (3) scenarios. However, no repression is present in the data, suggesting that the assumption of keeping K_r_ constant is essential in explaining all three constructs, especially the [100] construct. This result is further supported by the quantification of model performance given by the Akaike Information Criterion shown in Figure S16F. The Akaike Information Criterion is a metric that reports the performance of the model. Briefly, Akaike Information Criterion quantifies the deviation between the model and the data, then penalizes the model by the number of free parameters. As a result, the smaller the Akaike Information Criterion is, the better the model performs. Our analysis suggests that the assumption of K_r_ remaining constant and w_rp_ varying is favorable. Finally, the inferred parameters from the case where we let both K_r_ and w_rp_ vary across constructs are shown in Figure S16G, showing that the inferred K_r_ values are comparable between constructs (within error bars), whereas w_rp_ values vary.

2. A simple plot showing mRNA/cell as a function of repressor concentration for the different configurations in Figure 6 would be much appreciated. Perhaps I missed the inference, but the ultimate goal of an input/output understanding is to map concentration of a regulator to expression level of the target gene.

In the new Figure S18, we now present the accumulated mRNA as a function of repressor concentration for all of our synthetic enhancers. We would like to emphasize that this visualization could be misleading since Bicoid and Runt both vary along the anterior posterior axis of the embryo. Thus, to avoid confusion regarding the simultaneous variation of Bicoid and Runt across the embryo, in the main text we favor our current visualization where we plot our transcriptional output along the embryo’s body axis (anterior-posterior axis).

Reviewer #2 (Recommendations for the authors):The manuscript has several issues as noted below.1) For MCMC fitting of experimental data, the absolute error or relative error of all the results should be given. One would like to know the confidence degree of the fitting results.

Thank you for pointing this out. We agree that we missed this important information in our first submission. We have now included the 95% confidence interval for all our MCMC fits and prediction plots in our revised submission. As a result, Figures 4, 5, 6, and 7 now contain proper confidence intervals from the MCMC inference.

2) Formula (5) in the main text is used to fit the data, and the results are shown in figure6. However, the number of the parameters in the fitting formula is almost the same as the amount of the experimental data, and the formula is too complicated. The corresponding parameter space is quite large. Therefore, it is necessary to further analyze the accuracy of the fitting and the necessity of adding the cooperativity.

We agree that what is now Equation 6 is quite complicated and that we did not clearly present which parameters are free and which parameters are fixed because they were determined using previous measurements of reporter constructs with simpler regulatory architectures. We revised the way we display the formula to emphasize that only w_rr_ and w_rrp_ are the free parameters by coloring these two parameters in red. Thus, the number of free parameters is indeed smaller than the number of data points.

3) What is source of error bar of the initial RNAP loading rate? Need to give detailed explanation. If the source of error bar is from the data of multiple traces, why not fit multiple trajectories separately?

We thank the reviewer for pointing out this lack of clarification in our manuscript. Figure 3I shows how the initial rate of RNAP loading is measured by extracting the slope resulting from a linear fit to the averaged MS2 time traces at the beginning of nuclear cycle 14. Here, the MS2 traces were averaged over a set of nuclei in an individual embryo that are in the same spatial bin along the anterior-posterior axis of the embryo (2.5% of the embryo length) as previously described (Garcia et al., Current Biology 23:2140, 2013; Eck and Liu et al., *eLife* 9:e56429, 2020). Then, for all figures throughout this paper, we averaged these initial RNAP loading rates over multiple embryos (>3 embryos) at each spatial position, with an error bar corresponding to the standard error of the mean. Thus, the source of the error bar in the initial RNAP loading rate stems from embryo-to-embryo variability. This information has now been included throughout the paper, especially in Figure 3 caption.

To address the reviewer’s concern of whether our parameter inference would change if we inferred the model parameters from individual embryos, instead of the mean initial RNAP loading rates averaged over multiple embryos, we now included a new figure, Figure S20. In Figure S20B, we show that the initial RNAP loading rate obtained from individual embryos and then averaged over these embryos does not differ significantly from the initial RNAP loading rate calculated from an averaged embryo. Additionally, for a construct with the largest embryo-to-embryo variability, [010], we compared the results of performing MCMC inference using these two different approaches to calculating the initial RNAP loading rate. In Figure S20C, we now show that the values of parameters inferred using both approaches are comparable.

4) The fitting equations (1), (2) and (5), are all the fitting function for the experimental data. Are the parameter b, p and r the fitting parameters? If they are the fitting parameters, the fitting result needs to be given. These parameters are related to the weights of different binding states in your model, which need to be listed in detail, and compared the differences of fitting results under variant conditions.

We apologize for the lack of clarity in the presentation of these equations in our first draft. The parameters b, p, and r are b=[Bicoid]/K_b_, p=[RNAP]/K_p_, and r=[Runt]/K_r_, respectively. Thus, the fitting parameters are K_b_, p, and K_r_. However, K_b_ and p were already fitted in Figure 4 in the absence of Runt protein. These inferred parameters were taken as given in order to determine K_r_ in Figure 5. Thus, parameters are not inferred simultaneously, but sequentially. To clarify further, we now mark all free parameters in our equations in red in our manuscript to more easily distinguish free and fixed parameters.

5) In this paper, a variety of data is shown, which are Runt binding to different sites such as [001][010],[011] and so on. Please elaborate on how Runt is experimentally controlled or determined to bind to the specified sites.

We now elaborate on our synthetic enhancer construct design in Section 2.3 (Lines 219-226). Briefly, we used the same sequence motif that was identified in a study by Chen et al. (Cell 149:618, 2012). In that work, using synthetic enhancers, the binding motif used in our manuscript was validated to recruit Runt to repress the transcription of a *lacZ* reporter. Specifically, the resulting expression patterns were quantified using in situ hybridization in the presence and absence of Runt protein. We used the exact same sequence and positions of Runt binding sites across all our synthetic enhancer constructs, and we also observed transcriptional repression for all constructs by comparing the presence and absence of Runt protein. Additionally, this binding motif contains the canonical binding motifs for Run domain proteins, which have conserved amino acid sequences for the DNA binding domains between *Drosophila Run* and its human homologue *Runx1* (Chen et al., Cell 149:618, 2012; Koromila et al., Cell Reports 28:855, 2019). Several studies using ChIP-chip, or electric mobility shift assay (EMSA) have identified this consensus sequence to be directly bound by Runx1 protein’s DNA binding domain (Melnikova et al., Journal of Virology 67:2408, 1993; Lewis et al., Journal of Virology 73:5535, 1999). We have now included these references supporting Runt’s direct binding to its binding sites where we introduced our synthetic enhancer constructs in the main text.

6) The fluorescence of nucleus and cytoplasm is quantitatively analyzed, how to distinguish nucleus and cytoplasm? Please give the process and corresponding results.

We segmented nuclei by using the signal stemming from a histone fusion to infrared fluorescent protein (His-iRFP). We have included a new figure (Figure S22) to clearly show this. We have now clarified this point in the Materials and methods.

7) It is shown in Figure 6b that if the cooperativity is not included, a good fitting result cannot be obtained. Has the fitting covered a large enough parameter space search? The fitting results in figure 6 are quite different from the experiment results regardless of the cooperativity. If so, does it mean that the cooperativity plays a major role in the fitting, or even that the individual interaction between Runt and binding sites can be ignored? Could you separate out the part of cooperativity in the fitting results so as to confirm that the cooperativity plays a key role in the experiment?

We agree with the reviewer that this point has to be clarified further. In brief, the curves in Figure 6B did not have any free parameters, as we used the values for the parameters K_r_ and w_rp_ that were inferred from Figure 5 for each Runt binding site (w_rr_ = 1, and w_rrp_=1 in Equation 6 in this particular case).

To address the reviewer’s point about whether the higher-order cooperativity is sufficient to get a good fit without the interaction between Runt and its binding site, we performed a sensitivity analysis for the initial RNAP loading rate as a function of the Runt dissociation constant for its binding site, K_r_. Briefly, we fixed the K_r_ value to the inferred value from one-Runt binding site constructs shown in Figure 5, varied the K_r_ values by 100-fold (0.1x or 10x K_r_) and then repeated the MCMC inference for different scenarios of cooperativities to see if the higher-order cooperativity alone could give equivalently good fits to the data, regardless of the magnitude of K_r_. The result is shown in our new figure, Figure S21. Figure S21B-E, center column shows the case of a 10 times larger value of K_r_, where we found that the best MCMC fit cannot recapitulate our data even in the presence of both Runt-Runt cooperativity and higher-order cooperativity. On the other hand, Figure S21B-E, right column shows the case of K_r_/10, the best MCMC fit could recapitulate the data quite well if we invoked higher-order cooperativity, but ignored Runt-Runt pairwise cooperativity. In conclusion, we found that the Runt dissociation constant, K_r_, plays a pivotal role in fitting as well as the combination of different cooperativities, but that, even as we vary this dissociation constant, higher-order cooperativity needs to be invoked to obtain a quantitative agreement between theory and experiment.

Reviewer #3 (Recommendations for the authors):1) Results of Figure 5 should be accompanied by reports of fitting alternative models and suitable model comparisons performed. Also, a baseline model with constant K_R_ and \omega_RP_ (same for all three constructs) should be evaluated and compared to. Such comparison should ideally involve model comparison techniques such as likelihood ratio and AIC/BIC.

We have included a new figure, Figure S16 to address this point about alternative model assumptions in the context of the one-Runt binding site constructs. We tested alternative models with four different scenarios: (1) K_r_ and w_rp_ being constant across constructs, (2) K_r_ remains constant and w_rp_ varies, (3) K_r_ varies and w_rp_ remains constant, and (4) both K_r_ and w_rp_ vary across constructs. The results are given in Figure S16, where we show that our original assumption of keeping K_r_ constant and varying w_rp_ indeed explains the data better than the other models with the minimal set of free parameters. These results are further supported by a new calculation of the Akaike Information Criterion shown in Figure S16F.

2) Results of Figure 6 should be accompanied by statistical assessment of improvements due to higher order cooperativity parameters.

We again thank the reviewer for bringing up this point. We now include an Akaike Information Criterion (AIC) assessment for all models presented in Figure 6 (Figure 6G) to show how the model improves as we change our assumptions.

3) The fact that fitting of more complex models typically relied on setting different values for the additional parameters for each construct (rather than the same values for all tested constructs of that category) should be examined more closely. How well can the data be modeled without this flexibility?

Indeed, the simplest, baseline model would have been to consider equal values of additional parameters across the synthetic enhancer constructs such as when we introduce K_r_ and w_rp_ in Figure 5 to explain the repression by Runt protein for the one-Runt binding site cases. We considered the reviewer’s suggestion and performed MCMC inference keeping K_r_ and w_rp_ constant. However, the result was quite off from the experimental data as shown in Figure S16B.

4) The authors should address the concern that alternative model structures different from that used in the thermodynamics-based models here might have explained the data without requiring higher order cooperativity.

As the reviewer pointed out, our objective was to test whether thermodynamic models could predict Runt repressor action for a wide array of regulatory architectures. Within the thermodynamic modeling framework, we have tested different modes of repression, such as “competition” and “quenching'' as shown in Figure S7 and S8, and shown that they all require higher-order cooperativities in order to quantitatively recapitulate the data.

Of course, it is interesting to consider what would happen if we expanded our models to, for example, the non-equilibrium regime. While we are very interested in this approach, we believe that testing the non-equilibrium models is beyond the scope of this manuscript, and now address this point explicitly in the Discussion (Lines 491-499).

Other suggestions:One of the first experimental tests reported is that of predicted expression profile in the hunchback promoter with one Runt binding site. Figure 3J is supposed to be a qualitative match to the predictions in Figure 3F. Could the authors elaborate on what reasonable regimes of the model parameters would provide predictions that do not qualitatively match the observation? It seems like with an activator (Bicoid) progressively decreasing and a repressor (Runt) progressively increasing from anterior to posterior, the enhancer expression will obviously decrease in some form. Perhaps the authors might wish to make it clear that their goal here was to establish some basic parameter estimates to be used in the remaining analyses, and perhaps also to assess the uncertainties associated with those model parameters.

Figure 3J was given as an illustrative example of one of our measured profiles of the initial RNAP loading. Our objective with this figure was not to go into the detail of the reasonable parameter regimes that would be consistent with these data. Indeed, as the reviewer points out, the role of Figure 3J is to confirm the qualitative prediction that, given the Bicoid and Runt profiles, the initial rate of RNAP loading should decrease towards the posterior of the embryo. Figure 5 then attempts to quantitatively confront this type of data with our models to explore the reasonable regime of the model parameters as well as uncertainties. We clarified this in the main text, Section 2.2 (Lines 189-190) and Figure 3 caption.

The results presented in Figure 4 are puzzling. Figure 4D suggests that the differences among the constructs arises purely from differences in the p and R parameters, and not from the bicoid or bicoid-RNAP related parameters. Could the authors add a possible mechanistic speculation here regarding how making changes solely in the enhancer sequence might result in biochemical changes solely in the behavior of RNAP at the promoter (without any change in the protein-DNA interactions at the enhancer itself)?

Indeed, this result was puzzling to us. We think that our result reveals an important potential caveat in the study of synthetic enhancers: the basal level of expression should be measured experimentally in the absence of the transcription factor of interest. This can, of course, be experimentally challenging in cases where there are more than two transcription factors of interest.

As pointed out by the reviewer, the MCMC inference results shown in Figure 4 lead us to conclude that the different basal activities of our synthetic enhancer constructs in the absence of Runt protein is due to the variability in *p=[RNAP]/K_p_* and the maximum rate of transcription *R*. Currently, we do not have a satisfactory explanation for this observation. We would like to leave this for future investigations using experimental methods that can directly address these possibilities, but want to offer some speculations which we repeat from our response to Reviewer #1.

We speculate that, since the enhancer is proximal to the promoter, the overall enhancer sequence might affect the binding of the transcriptional machinery itself. Therefore, the transcriptional machinery might see slightly different DNA sequences, a combination of the promoter and part of synthetic enhancer for each construct. Indeed, Louder *et al.* (Nature 531:604, 2018), used cryo-EM to show that promoter-bound general transcription factor IID (TFIID) occupies roughly 60 bp of DNA (from -20 to +40, where 0 represents the start of the transcription site in the promoter) when it forms pre-initiation complex. Thus, it is conceivable that the general transcriptional machinery would interact with the general DNA sequence of an enhancer located proximally upstream from the promoter. In contrast, the binding site sequence for Runt is small enough (8 bp) compared to the enhancer sequence (~300 bp) that Runt protein might see a similar local DNA sequence for each binding site. Thus, our results could be explained if the general transcriptional machinery interacts with much longer DNA sequences than Runt repressor. This discussion is now included in the Discussion (Lines 433-440)

[Editors' note: further revisions were suggested prior to acceptance, as described below.]

The manuscript has been significantly improved but please further address one issue on the validation of transcription rate measurement-- the reviewers asked to use an orthogonal method such as smFISH to validate the measurement using MS2, which was not done in the revision. Please provide at least a thorough comparison and contrast of the current method with other orthogonal methods in the Discussion to equip readers with a full context of the quantifications that are central to the work.

Once again, we thank the editors and reviewers for their insightful comments and suggestions on our manuscript. We further address the issue raised by the editors on the validation of transcription rate measurement via comparison to orthogonal methods, such as smFISH below.

First, we quantified the accumulated mRNA level (pattern) by integrating the MS2 time traces during nuclear cycle 14. This method was previously validated to give quantitatively comparable results to single-molecule fluorescence in situ hybridization (smFISH) (Garcia et al., Current Biology 23:2140, 2013; Lammers et al., PNAS 17:836, 2020). Briefly, Garcia et al. compared the patterns of accumulated mRNA along the anterior-posterior axis of the embryo at late nuclear cycle 14 as quantified via MS2 and smFISH. These two methods yielded quantitatively comparable results as shown in Figure S3H from Garcia et al., Current Biology, 23:2140, 2013. We have elaborated further on this validation study in the Discussion (Lines 419-429).

Furthermore, we have compared two publicly available datasets from the same reporter construct driven by *hunchback* P2 enhancer by fluorescence in situ hybridization (FISH; Park et al., *eLife* 8: e41266, 2019) and MS2 ( Eck and Liu et al., *eLife* 9:e56429, 2020). Again, we integrated the MS2 time traces during nuclear cycle 13 for the datasets from Eck and Liu et al., and then compared that to the patterns of accumulated mRNA quantified via fluorescence in situ hybridization at early nuclear cycle 14 from Park et al. (shown in Author response image 2). By normalizing the accumulated mRNA profiles, we could see a good match along the anterior-posterior axis. From these comparisons to orthogonal methods widely used in the field (both smFISH and FISH), we claim that the MS2 technique is an accurate method for quantifying the patterns of accumulated mRNA as much as smFISH or FISH.

Note that the reviewer requested we repeat these experiments in the context of our work. However, we hope that the reviewers will agree that the already established relationship between the integral of the MS2 signal and the accumulated mRNA makes repeating these measurements unnecessary.

**Author response image 2. sa2fig2:** Comparison of accumulated mRNA profiles measured by FISH and MS2 for a reporter construct driven by the *hunchback* P2 enhancer. Normalized profiles of accumulated mRNA averaged over embryos from FISH (blue, Park et al., *eLife* 8:e41266, 2019) and MS2 (red, Eck and Liu et al., *eLife* 9:e56429). The blue and red curves show the mean and standard error of the mean over 16 and 9 embryos, respectively. The accumulated mRNA profiles were normalized to the average level within 25%-35% of the embryo length.